# Uncovering the mechanism for aggregation in repeat expanded RNA reveals a reentrant transition

Ofer Kimchi [1] ✉, Ella M. King [2] & Michael P. Brenner [2,3]

RNA molecules aggregate under certain conditions. The resulting condensates are implicated in human neurological disorders, and can potentially be designed towards specified bulk properties in vitro. However, the mechanism for aggregation—including how aggregation properties change with sequence and environmental conditions—remains poorly understood. To address this challenge, we introduce an analytical framework based on multimer enumeration. Our approach reveals the driving force for aggregation to be the increased configurational entropy associated with the multiplicity of ways to form bonds in the aggregate. Our model uncovers rich phase behavior, including a sequence-dependent reentrant phase transition, and repeat parity-dependent aggregation. We validate our results by comparison to a complete computational enumeration of the landscape, and to previously published molecular dynamics simulations. Our work unifies and extends published results, both explaining the behavior of CAG-repeat RNA aggregates implicated in Huntington's disease, and enabling the rational design of programmable RNA condensates.

RNA molecules form structures through base-pairing interactions between complementary regions. Frequently, a given region of an RNA molecule will be complementary both to another region on the same molecule as well as to a different RNA molecule. How is the competition between forming intra- and inter-molecular contacts decided?

Predicting the outcome of this competition is a major open question, affecting a wide swath of both in vivo and in vitro phenomena. The effects of this competition are particularly stark in the context of biological condensates, in which RNA–RNA interactions play a major, largely understudied, role[1–6]. While typical condensates often involve RNA–protein contacts, purely RNA-based aggregation phenomena have been observed both in vitro and in vivo for certain transcripts associated with repeat expansion disorders[7].

The expansion of repeats in certain sections of DNA has been implicated in a significant number of (primarily) neurodegenerative disorders including Huntington's disease, myotonic dystrophy, and Fragile X syndrome[8–10]. While the proximate cause of many of these disorders is the effect of the expansion on the protein sequence, these expansions can lead to effects at the level of the RNA as well[11–17], including an aggregation transition[7,18]. In particular, RNA containing **CAG** or **CUG** repeats were found by Jain & Vale to phase separate depending on the number of repeats present in each molecule, led by **GC** stickers binding to one another[7]. Since all **GC** stickers are self-complementary, it is not immediately clear what leads RNA molecules in certain parameter regimes to form inter- vs. intra-molecular contacts at different rates. Aggregation was observed when the number of repeats per strand exceeded ~30, roughly the same number of repeats leading to diseases in humans[7]. This phenomenon was also observed and further studied in molecular dynamics (MD) simulations of the system by Nguyen et al.[19]. These simulations were able to explore the molecular details of the aggregation transition, at the cost of each simulation (at a different concentration or number of repeats per strand) requiring ~3 months of supercomputer time.

[1]Lewis-Sigler Institute, Princeton University, Princeton, NJ 08544, USA. [2]Physics Department, Harvard University, Cambridge, MA 02138, USA. [3]School of Engineering and Applied Sciences, Harvard University, Cambridge, MA 02138, USA. ✉e-mail: okimchi@princeton.edu

Current models are insufficient to explore the properties of the aggregation transition demonstrated by these studies. State-of-the-art models of associative polymers either do not include a competition between intra- and inter-molecular binding (as is more natural for rigid proteins and for heterotypic interactions) or (erroneously) assume it has no qualitative effects on the resulting system[20–22]. While intra-chain interactions are typically ignored, exceptions do exist. These include Dobrynin's 2004 study extending the Flory–Stockmayer approach to include intra-chain associations[23], and a recent publication by Weiner et al. which found that self-bonds play a crucial role in determining phase behavior in a lattice system with heterotypic binding motifs of varying lengths[24].

Here, we derive an analytical model to describe a system of polymers with self-complementary stickers. Eschewing mean-field-theory approaches that have dominated the field, we employ a multimerization-based framework that predicts the entire multi-merization landscape in addition to the phase behavior, and thus naturally and explicitly considers the competition between intra- and inter-molecular contacts[25]. Quantitative consideration of this competition reveals that configurational entropy, arising from the multiplicity of ways to form bonds, is the driving force for aggregation in this system. Mapping out the complete phase diagram, we find that as a result of the competition between intra- and inter-molecular bonds, the system exhibits a tunable reentrant phase transition as a function of sequence or temperature. With very strong stickers (or low temperatures) the polymers fold into stable monomers and dimers, and are more likely to form aggregates at intermediate sticker strengths. We furthermore find that, for long enough linkers that enable adjacent stickers to bind, the parity of the number of stickers per strand affects not only the dimerization transition but the large-scale aggregation behavior as well. We validate our results by comparing them to a computational model that enumerates the complete landscape of intra- and inter-molecular structures that the RNA can form, and by comparing them to the results of the Jain & Vale and Nguyen et al.

studies[7,19]. Our work provides a unified framework to explain both dimerization and aggregation phenomena in **CAG** repeat systems[17,19] and extends these to arbitrary sequences, temperatures, and concentrations, thus setting the stage for the construction of novel materials and new techniques based on programmable RNA condensates.

## Results

### Equilibrium behavior is predicted by an analytical model

We consider a nucleic acid sequence comprised of $n$ identical stickers (Fig. 1a). The stickers are separated by $n-1$ equally spaced linkers that do not interact with the stickers. Each linker consists of $l$ nucleotides. Stickers are self-complementary and bind through base pairing interactions, such that each sticker can be bound to at most one other. Each bonded sticker has a free energy contribution of $F_b$; however, bonds that create closed loops also have an entropic cost $\Delta S_{loop}$ that depends on the loop length $l_{loop}$. This is because nucleotides comprising a closed loop (such as a hairpin, internal, or multi-loop) are constrained in the conformations they can adopt. A simple model treating unbound nucleotides as a polymer random walk estimates that the entropic cost of forming loops scales logarithmically with the loop length (see the "Methods" section)[26,27]. Assuming a characteristic loop length $l_{eff}$, the effective strength of the sticker interactions is $F \equiv F_b - T\Delta S_{loop}(l_{eff})$ (see the "Methods" section).

In this work, we are concerned with the behavior resulting from such sequences interacting with one another. Two stickers that bind to one another may be on the same strand or on two different strands. Moreover, many strands can be connected to one another through a chain of such bonds. We call a group of $m$ strands connected through a series of intermolecular bonds a multimer of size $m$, or an $m$-mer. There are many ways a multimer of size $m$ can form: any combination of bonds that occur either intra- or inter-molecularly within a group of $m$ strands, such that each strand is reachable from every other by following a series of intermolecular bonds, is an $m$-mer.

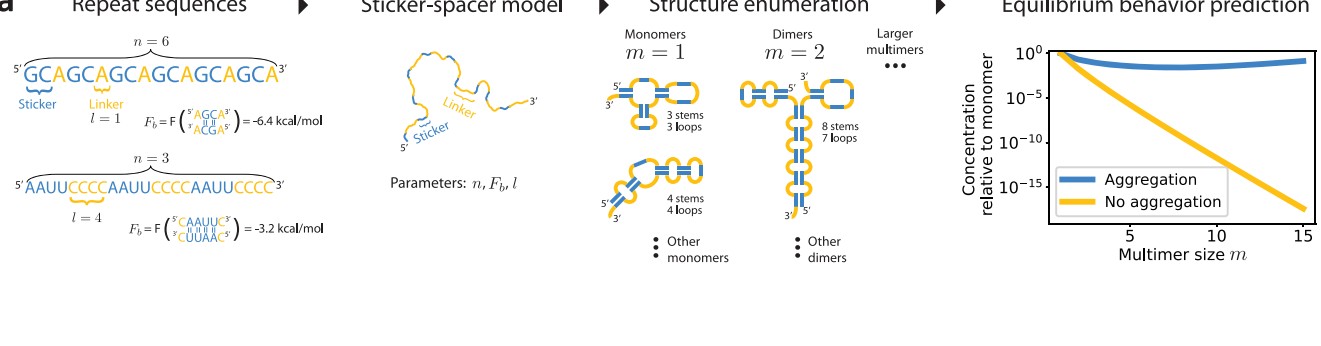

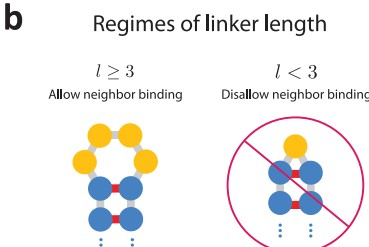

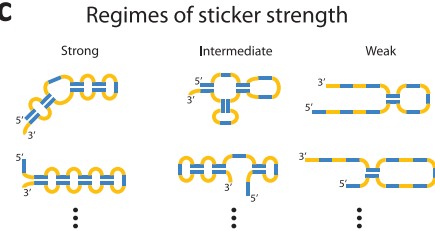

**Fig. 1 | Model overview. a** *Model procedure*: A repeat RNA or DNA sequence is converted to a sticker-spacer model, with stickers comprised of self-complementary regions. Possible structures, including multimers, are then enumerated by either computational or analytical methods. Partition functions are then calculated, leading to a complete description of the equilibrium behavior of the system, including the equilibrium concentrations of multimers. The system is in the aggregation regime when concentrations remain constant or increase with multimer size. **b** *Regimes of linker length*: The system can exhibit qualitatively different behavior depending on the length of the inert linkers. For long enough linkers, adjacent stickers can bind; for short linkers, they cannot because of hairpin size constraints. Structures visualized using *forna*[37]. **c** *Regimes of sticker strength*: For strong stickers, (almost) all of the sticker bonds are typically satisfied; for weak stickers almost none are; for intermediate strengths, the number of sticker bonds typically satisfied depends on a combination of the sticker strength and the multiplicity of structures in which a given number of stickers is bound.

We consider a system of $M$ strands present in a container of volume $V$, such that their concentration is $c^{\mathrm{tot}} = M/V$. We take the thermodynamic limit of $M$ and $V$ going towards infinity with their ratio staying constant. We seek to predict how frequently multimers comprised of $m$ strands form in this system, and how this frequency changes with $m$. We define $c_m$ as the concentration of multimers of size $m$, such that

$$c^{\mathrm{tot}} = \sum_{m=1}^{\infty} m c_m. \tag{1}$$

There are two possible regimes for the system: For large $m$, $c_m$ either decreases or increases with $m$ (Fig. 1a). In the former case, the system is in the dilute phase, with only small multimers typically forming. In contrast, if $c_m$ increases with $m$, large aggregates of the order of the system size dominate the landscape. The aggregation transition is defined as the crossover point between the regime in which very large multimers are suppressed, to that in which they are dominant.

In equilibrium, $c_m$ is proportional to the ratio of the partition function of $m$-mers, $Z_m$, to the partition function of $m$ monomers, $(Z_1)^m$ (see the "Methods" section). The partition functions are comprised of three terms:

$$Z_m = e^{-\beta(m-1)\Delta F} \sum_{N_b} g(n, m, N_b) e^{-\beta F N_b}. \tag{2}$$

Here, the multiplicity factor $g(n, m, N_b)$ represents the number of distinct ways to make $N_b$ bonds connecting $m$ identical strands, each with $n$ stickers. $\Delta F$ is the effective free energy cost of multimerization (see below) and $\beta = 1/k_B T$ is the inverse thermal energy, where $T$ is temperature. $g$ can be calculated exactly (see the "Methods" section and Supplementary Note 1) and is qualitatively different depending on whether the linkers are long enough to allow adjacent stickers to bind to one another or not (Fig. 1b).

In order to fit experimental data on the prevalence of multiple nucleic acid strands binding to one another in vitro, nucleic acid models include a free energy penalty for multimerization. This leads to the term $(m-1)\Delta F$ in Eq. 2. This penalty is motivated by the enthalpic and entropic costs of nucleic acids binding, including ion effects and the translational and orientational entropies lost upon association[28–30]. This penalty scales linearly with the number of strands in a multimer, such that each additional strand added to a multimer carries the same penalty[31]. See the "Methods" section and Supplementary Note 2 for further discussion.

The sum in Eq. 2 can be approximated by its dominant term (a saddlepoint approximation). There are three regimes to consider, corresponding to strong, intermediate, and weak binding, in which the sum in Eq. 2 is dominated by large, intermediate, and small values of $N_b$, respectively (Fig. 1c). The value of $N_b = N_b^*$ that dominates the sum is that which maximizes a combination of the bond energy $F$ and configurational entropy $g$. For example, the strong binding regime is characterized by bond energy considerations overwhelming configurational entropy effects, while the intermediate binding regime is characterized by a degree of balance between the two.

**The model is validated by comparing to exact computational enumeration and previously published results**

To validate the analytical model, we constructed a dynamic programming-based computational model that exactly enumerates $Z_m$ in polynomial time (Supplementary Note 5.2). The analytical model described above makes three primary approximations compared to the computational model: (1) it assumes a constant entropy for all loops; (2) it considers only structures with a given number of bonds $N_b$

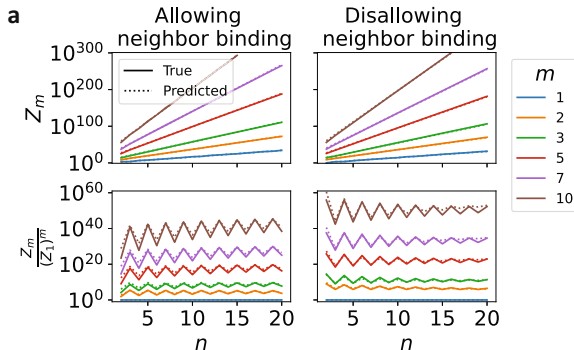

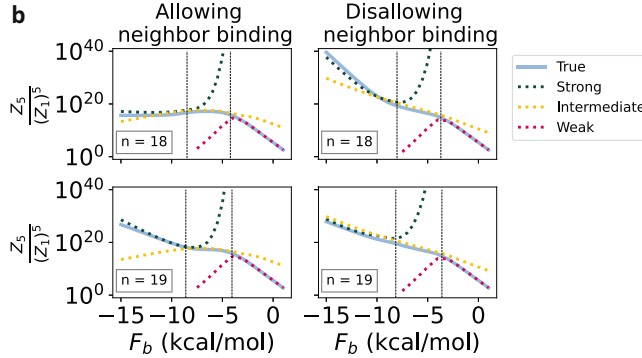

**Fig. 2 | Analytical model demonstrates good agreement with computational results. a** *As a function of number of stickers per strand*: Partition functions and partition function ratios are plotted with respect to $n$ using the exact computational (solid) and simplified analytical (dotted) models. A single fitting parameter was used for the analytical models, fit to the monomer partition function (top row, blue). The slight discrepancy in the analytical prediction for large $m$ and $n$ disallowing neighbor bonds is primarily due to the heuristic approximation of $g(n, m, N_b)$ from $g(nm, 1, N_b)$ used. **b** As a function of binding strength: The ratio of the pentamer partition function to that of five monomers is plotted; similar results hold for any other multimer chosen. The analytical model predictions are separated into three regimes: strong (green), intermediate (yellow), and weak (red) binding. Vertical dashed lines separate where different regimes are expected to provide the best agreement and are calculated as the values of $F_b$ such that $N_b^* = N_b^{\max} - 1$ and $N_b^* = N_b^{\min} + 3$. A single fitting parameter—the same one from panel (**a**)—is used.

(with a single next-order correction term); (3) it uses an approximate form for $g(n, m, N_b)$ (see the "Methods" section). The computational model makes none of these approximations, considering all (non-pseudoknotted; see Supplementary Note 5.1) structures that can form and including a loop-length-dependent loop entropy term.

Nevertheless, the analytical model closely approximates the exact computational model, as demonstrated in Fig. 2. The analytical model requires only one fitting parameter: the normalized effective loop length $l_{\mathrm{eff}}^{\mathrm{fit}}$ (see the "Methods" section). That parameter is fit separately to the regimes allowing and disallowing neighbor binding. Importantly, it is fit only once for each regime—to the monomer partition function with strong binding—and not separately for different values of $n$, $m$, or $F_b$. We demonstrate quantitative agreement between the analytical and computational models in Fig. 2, and in Supplementary Fig. 5.

We further sought to compare the model's predictions to previously published results, namely the MD simulations performed by Nguyen et al.[19]. Those simulations examined 64 **CAG**-repeat RNA strands with varying numbers of repeats per strand and of RNA concentrations. We considered the same system of **CAG** sequences, using the value $F_b = -10$ employed in the MD simulations and no fitting parameters beyond the aforementioned single parameter fit to the computational model. We enumerated the monomer and dimer

partition functions computationally, and used the analytical model to extrapolate up to $m = 64$, the number of strands used in the MD simulations. The extrapolation was performed by fitting the single parameter to our computational results for $m = 1$, and using

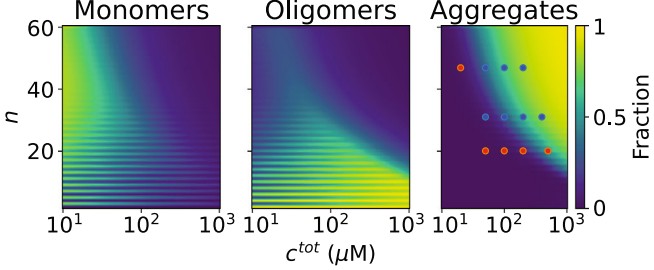

**Fig. 3 | Landscape of CAG repeats.** The equilibrium fraction of strands folded into monomers, oligomers (2–4-mers; primarily dimers), and aggregates are shown and compared to Nguyen et al.'s molecular dynamics (MD) simulation results. As the Nguyen et al. simulations used a sticker strength of $F_b = -10$ kcal/mol[19], we used the same sticker strength, with no fitting parameters to the simulations whatsoever. The MD simulation results are plotted as points in the aggregates panel, with blue points representing conditions for which aggregation was found, and red points for those in which it was not. We note that each of these points is a separate simulation taking 3 months of supercomputer time[19], in comparison to our analytical model for the entire landscape. In this system, neighbor binding is disallowed, monomers and dimers are in the strong binding regime, and multimers of $m \geq 3$ are in the intermediate regime. Aggregation is predicted for large concentrations and numbers of stickers per strand. Dimerization is less common as $n$ increases, while dominant for small values of $n$, especially odd values.

Supplementary Eqs. S38 and S48 to obtain the results for $m > 2$. The primary difference between our model predictions and those of MD simulations is that the former is purely equilibrium, while the latter is decidedly not so, even after significant simulation time. (A secondary difference is that the former considers an infinite system of given concentration, while the latter considers a finite number of strands).

We plot the propensity of the system to form aggregates as a function of $n$ and $c^{tot}$ in Fig. 3. Following ref. [19], we define multimers of size $2 \leq m \leq 4$ as oligomers; however, this ensemble is dominated by dimers, with trimers and tetramers forming at very low fractions. We find that for certain concentrations, the system forms either monomers or dimers depending on the parity of $n$, in agreement with experimental results[17]; however, this parity does not significantly affect aggregation. We plot the results of Nguyen et al. on top of our predictions as colored points, finding excellent quantitative agreement between the two.

**A reentrant phase transition governs aggregation as a function of sticker strength**

For very low temperatures or strong stickers, the ensemble of multimers is dominated by small structures such as dimers, in which all bonds can be satisfied. However, for intermediate sticker strengths, the configurational entropy gain of having a few unsatisfied bonds exceeds the energetic cost. This configurational entropy grows with multimer size, driving the system to aggregate. Finally, for very weak stickers or high temperatures, the structures melt. This phenomenon corresponds to a reentrant phase transition. We demonstrate this transition in our computational model in Fig. 4, enumerating up to $m = 15$. As shown in the figure, the two dilute phases at strong and weak

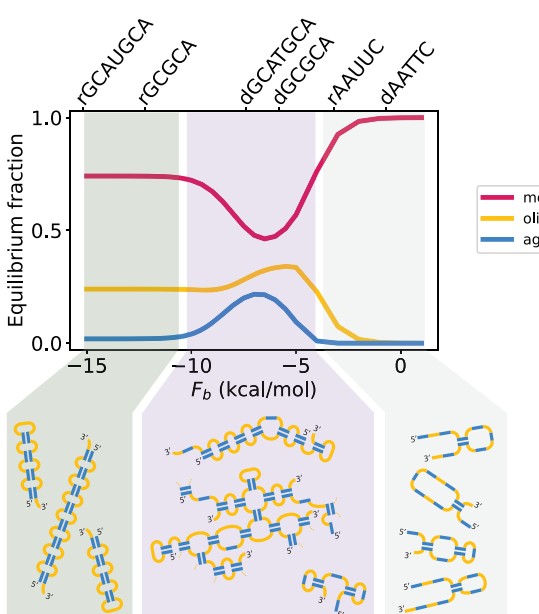

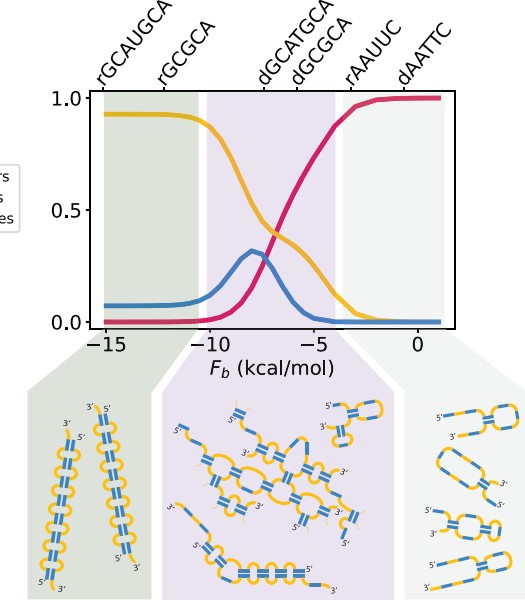

**Fig. 4 | A reentrant transition as a function of sticker strength.** Enumerating the exact partition functions up to $m = 15$ with the computational model, we find a reentrant transition with respect to $F_b$ in both the regime allowing neighbor binding (panel **a**; $n = 8$, $l = 4$, $c^{tot} = 8$ mM is shown) and the regime disallowing neighbor binding (panel **b**; $n = 8$, $l = 1$, $c^{tot} = 4$ mM is shown). The high concentration used is a result of the lack of $Mg^{2+}$ considered explicitly in the model; see the "Discussion" section. Aggregates (defined as $m \geq 5$-mers in accordance with ref. [19]) are most likely to form for intermediate sticker strengths, since very strong stickers lead to stable monomers (red) or dimers (dimers, trimers, and tetramers comprise the orange curve). Although aggregates are suppressed in both strong (green background; left) and weak (gray background; right) binding regimes, the molecular structures

of monomers and dimers in these regimes are quite different: in the former, all or nearly all bonds are satisfied in a typical molecule, while very few bonds are typically satisfied in the latter regime. For this reason, the strong binding regime of the short linker case (i.e. disallowing neighbor binding) is predicted to contain a large concentration of dimers (which can satisfy all sticker bonds), and few monomers (which cannot). In the long linker case (i.e. allowing neighbor binding), for even values of $n$, monomers are also able to satisfy all bonds and are thus present at high concentrations in the strong binding regime. Top axis shows example sequences for RNA (r) and DNA (d), and their sticker strengths as calculated by the nearest-neighbor model, enabling a direct match from sequence to model predictions[29,30].

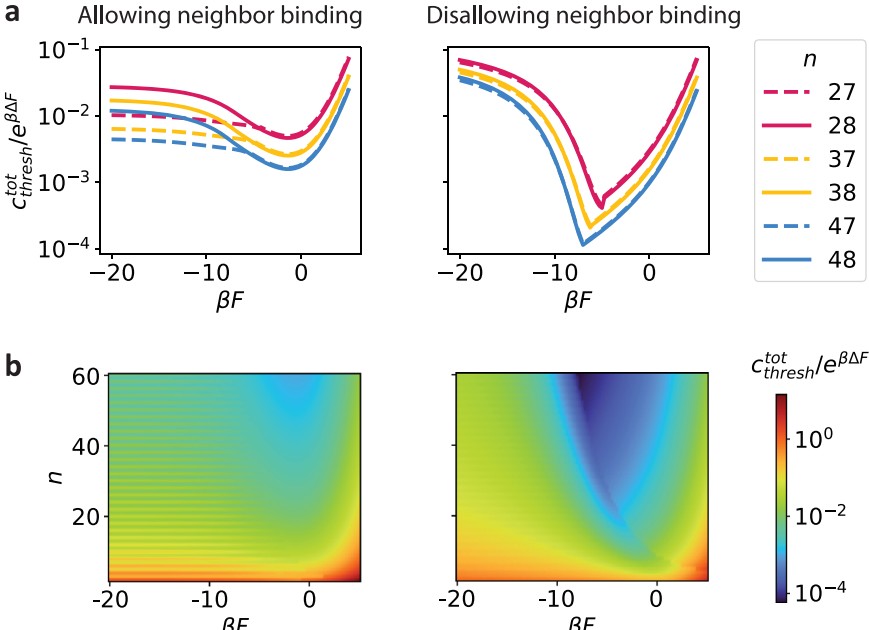

**Fig. 5 | Phase diagram. a** *Reentrant transition in the analytical model*: The analytical model enables enumeration up to arbitrarily large $m$, and reveals a reentrant transition. With high enough concentrations of RNA, aggregation is always possible; however, for certain concentrations, the analytical model predicts the system will undergo a reentrant phase transition in agreement with computational results (Fig. 4). Panel **a** shows slices for certain values of $n$ through the complete phase diagram shown in panel (**b**). Parity of $n$ affects aggregation phenomena for the system allowing neighbor binding (LHS). **b** *Complete phase diagram*: The complete phase diagram as predicted by enumeration up to arbitrarily large $m$ with the analytical model is displayed. The normalized concentration needed to achieve aggregation is displayed as a function of $n$ and $\beta F$. The reentrant transition is especially apparent for short linkers (RHS) as well as for long linkers with even values of $n$ (LHS). Systems with long linkers typically require higher concentrations to aggregate than those with short linkers, since monomers are typically more stable in the former case. Discontinuities are due to the model's approximation of an abrupt transition from the strong to intermediate binding regimes for monomers. $c_{thresh}^{tot}$ is made dimensionless by dividing by the multimerization cost $e^{\beta\Delta F}$ (see Supplementary Note 2).

binding regimes are quite different from one another. In the strong binding regime, (almost) all bonds are satisfied in a typical structure, mainly through intramolecular interactions or dimerization. In the weak binding regime, (almost) no bonds are typically satisfied.

We next explored whether this reentrant transition was merely a small $m$ effect. We employed the analytical model, for which we can consider arbitrarily large values of $m$. Even when considering $m \to \infty$, we find a reentrant transition in the threshold concentration above which the system is expected to form aggregates, $c_{thresh}^{tot}$ (see Supplementary Note 4), as shown in Fig. 5a. This transition is especially prominent for short linkers that disallow neighbor binding since the configurational entropy of dimers in this regime is quite limited (regardless of $n$, only one dimer configuration can satisfy all stickers). For longer linkers (allowing neighbor binding), this transition is most pronounced for even values of $n$ for which monomers can satisfy all their own bonds, although it is apparent also for odd $n$, for which dimers can satisfy all bonds.

The behavior shown in Fig. 5 is in agreement with what we would expect from configurational entropy concerns alone (Supplementary Fig. 6). That the propensity of the system to aggregate occurs at more negative values of $\beta F$, and is more pronounced, for the case of disallowing neighbor binding than for the case of allowing neighbor binding, is predicted by the different forms of the configurational entropy in these two regimes. Similarly, larger values of $n$ increase the propensity of the system to aggregate because of their effect on configurational entropy, rather than any enthalpic considerations (Supplementary Fig. 6).

## Discussion
In this work, we have considered a simple model of competition between intra- and inter-molecular binding: a polymer with $n$ identical evenly spaced self-complementary stickers. We have shown that the

system is characterized by three parameters: $n$, the number of repeats per strand; $\beta F$, the effective strength of each bond accounting for the loop entropy cost; and $c^{tot}e^{-\beta\Delta F}$, a dimensionless concentration that accounts for multimerization cost.

Our model computes the prevalence of all possible multimers that can form, considering both intra-strand and inter-strand contacts. Our framework quantitatively recapitulates previously published MD simulation results, each data point of which required 3 months of supercomputer simulation time[19]. We substantially extend these results to arbitrary sequences, temperatures, and concentrations, and to arbitrarily large multimers (i.e. aggregates) in an analytical framework.

In this system, aggregation is not necessarily predicted as the regime where the most possible bonds are satisfied, as bonds can be satisfied by intramolecular as well as by intermolecular contacts. Instead, aggregation is predicted by the relative stability of the aggregate compared to smaller multimers. The stability of each structure is a function of three terms, as seen in Eq. 2: (1) the number of stickers bound (each contributes $F$ to the free energy); (2) the number of strands in the structure (each contributes $\mu + \Delta F$, where $\mu$ is the chemical potential); and (3) the configurational entropy of the structure. This last term contributes $-\log(g)/\beta$ to the free energy, where $g$ is the number of ways to satisfy the given number of bonds with the given number of strands in the structure.

This last term is the driving force for aggregation in this system. Aggregates are no more stable than dimers in terms of the first term, the possible number of stickers bound (both are able to satisfy all stickers). Aggregates are further penalized by the second term, the multimerization cost. If these two terms were the only terms in the free energy, we would not see any aggregates. It is the third term, the configurational entropy, that drives aggregation. Larger multimers are able to satisfy their bonds in many more configurations than a corresponding collection of smaller multimers, leading to an enormous

entropic benefit in forming aggregates. This has been described as a competition between configurational and translational entropies in other contexts[24,32]. In our system, the benefit due to $g$ peaks when most, but not all, stickers are satisfied (Supplementary Fig. 6).

This behavior leads to a reentrant phase transition. For $-\beta F \gg 1$, the number of bonds satisfied is the primary consideration. Dimers are able to satisfy all their bonds, and the multiplicity benefit of aggregates is not sufficiently large when all bonds are satisfied, suppressing aggregation in this regime. Aggregation is also suppressed for very positive values of $\beta F$, which as a result of loop entropy costs can occur even when the sticker binding itself is favored (i.e. $F_b < 0$). However, for intermediate values of $\beta F$—when dimers prefer having some bonds left unsatisfied—the configurational entropy benefit of forming aggregates is overwhelming. Aggregates form at 1–2 orders of magnitude lower concentrations in this regime than in the strong binding regime.

The predicted aggregation transition of the system is completely described in Fig. 5b. We plot the (dimensionless) threshold concentration $c_{thresh}^{tot}$ as a function of $n$ and $\beta F$. Aggregation is more prevalent for short linkers (disallowing neighbor binding) than for longer linkers (allowing neighbor binding). For short linkers, small structures are quite constrained in the number of ways they can satisfy all of their bonds, leading the differential configurational entropy benefit of aggregates to grow quite large. For longer linkers, smaller structures are more stable since the corresponding multiplicity is much larger. For similar reasons, the reentrant phase transition is most pronounced with short linkers. For long linkers, even values of $n$ demonstrate a more pronounced reentrant transition than odd values, since their competition is between monomers—with no multimerization penalties —and aggregates. In all other cases, the reentrant transition is primarily due to competition between dimers and aggregates. For short linkers, the parity of $n$ is found in our model to affect monomerization vs. dimerization in agreement with previously published results[17], but has almost no effect on aggregation properties. The reason is that for short linkers and strong stickers, dimers behave similarly regardless of the parity of $n$: both odd and even $n$ can form a dimer satisfying all bonds with only one configuration.

Although there is a qualitative difference between short linkers of $l < 3$ and long linkers of $l \geq 3$, within each regime, increasing the linker length leads to larger values of $\Delta S$ and weaker binding. Decreasing the persistence length, for example by changing ionic conditions, would be expected to lead to a similar result. These effects and the predicted phase diagram as a whole (Fig. 5b) could be at least qualitatively tested experimentally by replicating the Jain & Vale experiments for multiple sequences with different sticker strengths and linker lengths and measuring the change in the concentration needed to form aggregates for the different conditions. The available published data is in good agreement with our predictions, in that larger values of $n$ show a greater propensity for aggregation in both experiments and our model predictions[7].

Our results bear similarities to the so-called "magic number effect" whereby for heterotypic mixtures, aggregation is suppressed when the number of binding sites in one species is a small integer multiple of the other's[32,33]. In such systems, small stable clusters can form with all bonds satisfied. In our homotypic system, dimers can always exhibit a magic number-like effect for strong stickers, and in the regime in which neighbor binding is allowed, for even $n$, monomers can as well. In fact, a weak reentrant transition has been observed in some simulations of the magic number effect in heterotypic systems (see Fig. 3A of ref. [34]). Our results suggest that a reentrant transition may be a generic feature of the magic number effect and that the strength of the reentrant behavior may decay the more molecules are involved.

Our model has several limitations. To make the expression analytically tractable, our formalism makes a heuristic approximation for the multimer multiplicity factor $g$ in the regime disallowing neighbor

bonds. For similar reasons, we were unable to analytically explore the weak binding regime, applicable for systems where the loop entropy cost of forming stickers outweighs their energetic benefit. A limitation of our model's physiological applicability is that we did not explicitly consider magnesium. Magnesium can act as a bridge between negatively charged RNA molecules such that even in the absence of base pairing, Mg–RNA mixtures can form aggregates[18,35]. Experimental results thus rely on magnesium aiding the aggregation process[7]. However, the MD simulations to which we compare here do not explicitly consider magnesium[19] and the high concentrations required for the system to aggregate (e.g. Fig. 3) are the result. To first-order, the effects of magnesium could be accounted for in our model as modifying $\Delta F$ (along with $F_b$), which effectively modifies the concentrations, as concentrations only enter the model as $c^{tot}e^{-\beta\Delta F}$. For clarity, we opted to leave $\Delta F$ unmodified; therefore, the high concentrations we consider should be significantly decreased for a system including magnesium.

While non-equilibrium effects are relevant in these systems, our analysis is entirely an equilibrium prediction. Indeed, kinetic trapping appears to be the biggest experimental hurdle to testing our reentrant phase predictions. At the same time, the results of decidedly out-of-equilibrium MD simulations[19] show excellent quantitative agreement with our equilibrium predictions (Fig. 3). For this reason, it is likely that out-of-equilibrium effects are not the dominant factor in repeat RNA aggregation behavior. In vivo RNA aggregates are even more fluid-like and dynamic than in vitro aggregates, for reasons that remain largely unclear but appear to be the result of active enzymes in the cell[7]. Future work may consider how such active processes affect the aggregation properties, and the connection between in vivo non-equilibrium steady states and the equilibrium steady state discussed here.

Given the radical simplicity of the model used here, there is a host of extensions to consider. For example: How does this model interact with complex coacervation, as when including polycations in the solution? How does a polymer pattern with multiple orthogonal stickers behave? How do multiple different polymers, with both *cis* and *trans* binding, interact with one another? And how do physiological RNA molecules use the principles explored here to control their aggregation properties?

Our work demonstrates that the competition between intra- and inter-molecular binding can lead to remarkable and (perhaps) unintuitive behavior. Our results mapping the control knobs for this phase behavior create a framework for the study of RNA–RNA interactions in in vivo biological condensates and set the stage for the construction of novel materials and new techniques based on programmable RNA condensates.

## Methods

### Partition functions determine equilibrium behavior

We consider a nucleic acid sequence comprised of $n$ stickers separated by $n-1$ linkers (Fig. 1a). Stickers are self-complementary and bind through base pairing interactions, such that each sticker can be bound to at most one other sticker. The strength of the sticker interactions, $F_b$, is determined by the sequence of the stickers; for example, an RNA **GC** sticker with **A** nucleotide linkers in standard conditions has $F_b = -6.4$ kcal/mol (or, for DNA, $-1.4$), while a **GCGC** sticker has $F_b = -12.2$ kcal/mol ($-5.8$ for DNA). These are calculated using the classic nearest-neighbor model for RNA or DNA base-pairing interactions[29,30]. The linkers, each of which is of length $l$, are inert.

We seek to predict how frequently multimers comprised of $m$ strands form, and how this frequency changes with $m$. Aggregation occurs in the parameter regime where the concentration of multimers comprised of $m$ strands, $c_m$, increases with $m$. $c_m$ is defined as the sum of all structures that have $m$ strands connected by base pairing interactions. In equilibrium, $c_m$ is proportional to the partition function of

$m$-mers, $Z_m$:

$$Z_m = \sum_{\sigma_m} e^{-\beta F(\sigma_m)}. \tag{3}$$

Here, $\sigma_m$ is a structure comprised of $m$ strands linked by base pairing, including potential intramolecular bonds; and $\beta = 1/k_B T$ where $k_B$ is Boltzmann's constant and $T$ is the temperature measured in Kelvin. $F(\sigma_m)$ is the free energy of the structure, given by[29]

$$F(\sigma_m) = F_b N_b(\sigma_m) + (m-1)\Delta G_{assoc} - T \sum_{loops} \Delta S_{loop}(l_{loop}), \tag{4}$$

where $N_b(\sigma_m)$ is the number of bonds in the structure, and $\Delta G_{assoc}$ is the hybridization penalty associated with intermolecular binding (discussed below). Each closed loop of length $l_{loop}$ leads to an entropic penalty of $\Delta S_{loop}(l_{loop})$, associated with the decrease in three-dimensional configurations of the single-stranded region of the loop compared to a free chain, given by[26,27]

$$\Delta S_{loop}(l_{loop}) = k_B \left[ \ln v_s + \frac{3}{2} \ln \left( \frac{3}{2\pi b \, l_{loop}} \right) \right], \tag{5}$$

where $v_s$ is the volume within which two nucleotides can bind, and $b$ is the persistence length of single-stranded regions. This equation treats the single-stranded loop as an ideal chain. An excluded volume term $v m^2$ can be added to Eq. 4[20] but we assume $v$ is small enough that this term is negligible except for very large $m$ (see Supplementary Note 4 for further discussion).

Given the partition functions $Z_m$ for all $m$-mers, we can calculate the equilibrium concentrations of $m$-mers, $c_m$, for all $m$, by solving a set of $m$ simultaneous equations. $Z_m$ affects physical observables such as $c_m$ only through the ratio $Z_m/Z_1^m$, describing, in essence, the propensity of $m$ strands to form an $m$-mer as opposed to $m$ monomers[25,31]:

$$c_m = \frac{Z_m}{Z_1^m} c_1^m$$
$$\sum_m m c_m = c^{tot} \tag{6}$$

where the concentrations are made dimensionless by normalizing by a reference concentration (see Supplementary Note 2) and $c^{tot}$ is the total concentration of strands added to solution. In short, this equation arises from $c_m = Z_m e^{m\beta\mu}$ where $\mu$ is the chemical potential and the fugacity $e^{\beta\mu} = c_1/Z_1$ in equilibrium[25].

Solutions to Eq. 6 have two typical regimes. In one, $c_m$ decays exponentially with $m$. On the other, $c_m$ grows with $m$ (until excluded volume effects begin to dominate). The latter regime corresponds to aggregation (Fig. 1a).

**An analytical model for the partition functions**
The calculation of $Z_m$ is too computationally intensive to perform directly, by explicitly enumerating all possible structures that can form, as the number of possible structures grows exponentially with $n$ and $m$. In order to predict phase behavior for a wide range of sequences and experimental conditions, we develop an analytical framework for computing $Z_m$. This framework enables us to search a broad parameter space and tune phase behavior in the system. We validate our analytical model against a computational model that exactly calculates $Z_m$ with a dynamic programming approach (Supplementary Note 5.2) thus providing an exact baseline model for comparison.

We rely on one major assumption to enable an analytical approach: we approximate the loop entropies as independent of loop length; or equivalently, we assume that the model is dominated by loops of one characteristic length, $l_{eff}$. This length depends on the length of the linkers in the system, $l$. This approximation is reasonable because of two factors. First, because of the logarithmic dependence of $\Delta S_{loop}$ on loop length (Eq. 5), moderate heterogeneities in loop length lead to only small differences in $\Delta S_{loop}$. Second, because the typical number of loops in a multimer scales linearly with the size of the multimer (see Supplementary Note 3), we expect similar levels of heterogeneity in loop length independent of the size of the multimer. This approximation is expected to break down for very large $n$ and weak binding ($F_b > 0$), in which case the few loops that typically form will likely have a broad distribution of lengths; this regime is not considered here.

With this approximation, for monomers, each bond provides constant free energy of $F = F_b - T\Delta S$, where $\Delta S = \Delta S_{loop}(l_{eff})$. Since the number of loops is given by $N_b - (m-1)$, we also define $\Delta F \equiv (\Delta G_{assoc} + T\Delta S)$. This quantity enters Eq. 6, such that it allows us to redefine a rescaled concentration $c e^{-\beta\Delta F}$ (also, see Supplementary Note 2). Without rescaling concentration, the partition function $Z_m$ can thus be written as

$$Z_m = e^{-\beta(m-1)\Delta F} \sum_{\sigma_m} e^{-\beta F N_b(\sigma_m)}$$
$$= e^{-\beta(m-1)\Delta F} \sum_{N_b} g(n, m, N_b) e^{-\beta F N_b} \tag{7}$$

where the multiplicity factor $g(n, m, N_b)$ represents the number of distinct ways to make $N_b$ bonds connecting $m$ identical strands, each with $n$ stickers. This is identical to Eq. 2.

This multiplicity factor is most straightforward to consider for the case of monomers. We make the approximation that the contribution of pseudoknots to the partition function is negligible due to their high entropic cost (see Supplementary Note 5.1). Our goal is therefore to calculate the number of ways to form non-pseudoknotted structures containing $N_b$ bonds given a strand of $n$ stickers. For monomers, the multiplicity can be calculated exactly. However, the result depends on whether adjacent stickers are able to bind to one another or not. For a long enough linker length (≥3 nts for the case of RNA), neighboring stickers can bind; for shorter linker lengths (as, for example, for **CAG** repeats), they cannot (see Fig. 1b). As derived in Supplementary Note 1.1,

$$g(n, 1, N_b) = \begin{cases} \frac{n!}{(n-2N_b)!(N_b+1)!N_b!} & \text{if adjacent stickers can bind} \\ \frac{(n-N_b)!(n-N_b-1)!}{(n-2N_b)!(n-2N_b-1)!(N_b+1)!N_b!} & \text{otherwise} \end{cases} \tag{8}$$

The top line (allowing neighbor binding) is simply calculated as the product of two factors: $\binom{n}{2N_b}$ (the number of ways to choose $2N_b$ bound stickers from $n$ possibilities); and the Catalan number $C_{N_b}$ (the number of non-pseudoknotted ways to construct bonds between the chosen stickers). The bottom line (disallowing neighbor bonds) requires a brief additional calculation to derive (Supplementary Note 1.1).

Calculating $g(n, m, N_b)$ from $g(n, 1, N_b)$ also depends on whether or not adjacent stickers can bind (see Supplementary Note 1.2). While the exact calculation requires large numbers of sums with no closed-form solution, a close approximation is given by

$$g(n, m, N_b) \approx \begin{cases} \frac{g(nm, 1, N_b)}{m} & \text{if adjacent stickers can bind} \\ \frac{g(nm+\alpha(m-1), 1, N_b)}{m} & \text{otherwise} \end{cases} \tag{9}$$

where $\alpha \approx 0.42$, representing an additional heuristic for the case of disallowing neighbor binding compared to the case of allowing such binding. The value of $\alpha = 0.42$ used is a heuristic estimate that is an

especially good fit to the strong interaction regime, and other approximations may improve it (see Supplementary Fig. 1). The factor of $1/m$ corrects for overcounting due to symmetry (Supplementary Note 1.2.3; see also Supplementary Fig. 2)[36].

Given expressions for the multiplicity factor, the partition functions (Eq. 7) are now in principle computable. However, the full sum in that equation remains too computationally intensive to be useful. We, therefore, turn to a saddlepoint approximation: sums of exponentials are typically dominated by their maximum terms, and Eq. 7 is no exception.

In order to find the maximum term, there are three cases to consider, corresponding to physically meaningful distinctions (Fig. 1c). In one regime, the "strong binding" regime, the ensemble is dominated by structures that maximize the bond energy, and the sum is dominated by the last terms ($N_b = N_b^{max}$). In the second, the "intermediate binding" regime, the ensemble is dominated by structures that maximize a combination of the bond energy and configurational entropy measured by $g$, and the sum is dominated by an intermediate-term ($N_b = N_b^\star$). In the third, the "weak binding" regime, the ensemble is dominated by structures that have almost no bonds, and the sum is dominated by the first terms ($N_b = N_b^{min}$). These three cases must be treated separately: in the strong and weak binding regimes, the discrete nature of the sum is crucial, while in the intermediate regime, the sum can be well-approximated by an integral. The boundary between these regimes occurs approximately when $N_b^\star = N_b^{max} - 1$ or $N_b^\star = N_b^{min} + 3$. For Figs. 3 and 5, we set the boundary between the strong and intermediate regimes at $N_b^\star = N_b^{max} - \frac{1}{4}$ (allowing neighbor binding) and $N_b^\star = \frac{n}{2} - 2$ (disallowing neighbor binding).

After computing the dominant term of the sum, the next-order correction to $Z_m$ comes from either considering the next-dominant term (strong and weak regimes) or the curvature at the maximum (intermediate regime); see Supplementary Note 3 for more details.

When comparing between the analytical and computational models, we use a single fitting parameter $l_{eff}^{fit}$, which tunes the normalized effective loop length. That parameter is fit separately to the monomer partition functions allowing and disallowing neighbor binding, but is kept constant for all values of $m$. For different binding strengths, a different fraction of stickers will be bonded, leading to a different value of $l_{eff}$. Rather than having a separate fitting parameter for each parameter set, we only fit once (to monomers) in each of the two linker length regimes (allowing and disallowing neighbor binding). We then assume that $l_{eff}$ changes linearly with the fraction of stickers bonded, leading to:

$$l_{eff} = \frac{nm}{2N_b^\star} l_{eff}^{fit}. \tag{10}$$

We fit $l_{eff}^{fit}$ to the strong binding regime (Fig. 2) for which $l_{eff} \approx l_{eff}^{fit}$. We find intuitively reasonable values for $l_{eff}^{fit}$. When using $l = 1$ (disallowing neighbor binding), we find $l_{eff}^{fit} = 4.3$ nucleotides. This value is in between the length of an internal loop formed by two individual linkers (4 nucleotides) and the length of a hairpin loop formed by two linkers and a sticker (5 nucleotides). When using $l = 4$ (allowing neighbor binding), we find $l_{eff}^{fit} = 7$ nucleotides. This value is also in between the length of an internal loop formed by two individual linkers (10 nucleotides) and the length of a hairpin loop formed by a single linker (5 nucleotides).

### Reporting summary
Further information on research design is available in the Nature Portfolio Reporting Summary linked to this article.

## Data availability
The data that support the findings of this study are available from the corresponding author upon reasonable request.

## Code availability
All code used to generate the results and figures in this study can be found at https://github.com/ofer-kimchi/RNA-aggregation.

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

## Acknowledgements

We thank Sumit Majumder and Ankur Jain for sharing their expertise of this system and the central experiments, as well as Hung Nguyen and Naoto Hori for discussions of their molecular dynamics simulations. We thank Ned Wingreen and Yaojun Zhang for discussions of magic number systems and their connection to the present work. We also thank Krishna Shrinivas, Peter Clote, Megan Engel, Ben Weiner, and Rees Garmann for interesting and useful discussions. This work was supported by the Peter B. Lewis '55 Lewis-Sigler Institute/Genomics Fund through the Lewis-Sigler Institute of Integrative Genomics at Princeton University, and the National Science Foundation through the Center for the Physics of Biological Function (PHY-1734030) (O.K.); a National Science Foundation Graduate Research Fellowship under Grant No. DGE1745303 (E.M.K.); the Harvard Materials Research Science and Engineering Center (DMR 20-11754), the Office of Naval Research (ONR N00014-17-1-3029), and the Simons Foundation through the Simons Foundation Investigator Award (M.P.B.).

## Author contributions

All the authors (O.K., E.M.K., M.P.B.) designed research. O.K. and E.M.K. carried out theoretical calculations, wrote Python code, and analyzed data. All the authors (O.K., E.M.K., M.P.B.) wrote the article.

## Competing interests

The authors declare no competing interests.
