## [Peer Review File · Nature Communications]

Uncovering the mechanism for aggregation in repeat expanded RNA reveals a reentrant transitionREVIEWER COMMENTS

Reviewer #1 (Remarks to the Author):

Summary

Motivated by prior experimental and simulation studies that show that repetitive RNA sequences can exhibit gelation, the authors of this study have developed an analytical framework to understand the effect of linkers, interaction strength, and number of stickers on the aggregation behavior of associative polymers. The authors explicitly make a distinction between intermolecular and intramolecular bonds between the polymer molecules and take into account the corresponding energy and entropy effects. While this concept has been explored before, the authors adopt a new approach where they analytically compute partition functions of molecular clusters of each size, which can then be used to compute the concentration of clusters of each size. They define aggregation as the situation where the concentration increases with cluster size, where large clusters have the highest concentration. This is a nicer criterion than the Flory-Stockmayer criterion commonly used in associative polymer theory to define the aggregation threshold. Using this criterion, they define threshold concentrations for aggregation and study the impact of varying the interaction strength, number of stickers, and linker length. The paper presents a phase diagram and has some interesting insights. Most interestingly, the authors discover a re-entrant behavior in the aggregation behavior, where aggregation initially increases and then decreases upon increasing the binding strength of the stickers. Their model reveals the underlying physical mechanism. The authors' analytical framework can be helpful for understanding mechanisms, and it obviates the need for very expensive MD simulations. Therefore, it is suitable for publication in Nature Comms. However, a number of points need to be addressed before publication can be recommended

Specific comments:

1] In 2004, Dobrynin was the first to build a model for associative polymer aggregation moving beyond the mean-field approaches. Competition between intra- and inter-molecular bonds was also explicitly accounted for in this paper. However, Dobrynin mainly looked at the weak/intermediate binding regime and showed that taking into account this competition leads to a change in the boundary of the aggregation line in the temperature-volume fraction space. It is important to cite this paper.

2] The discussion of ΔG_{assoc} and ΔF , both in the main text and S2 is unclear. Eq. 6 seems correct if F is properly calculated. Why do we need the extra factors involving ΔF and ΔG_{assoc} ? This point should be more clearly described.

3] Further justification is needed for the use of the maximum term approximation, which implies that the typical value thus obtained is dominant.

4] In the discussion, the authors state the results from Ankur Jain and Ron Vale's paper¹ that in vivo RNA aggregates are more fluid-like and this is likely a consequence of active processes. They use this to claim that "Thus, our equilibrium results may be even more relevant in vivo than in vitro". The authors are inherently assuming that active processes that contribute to maintaining fluidity affect only the dynamics. In principle, these active processes can lead to non-equilibrium steady states that may be different from the equilibrium state.

5] When heuristically computing $g(n,m,N_b)$ for multimers in the case of short linkers where neighbor binding is not allowed, $g(n,m,N_b)$ was approximated as $g(mn+\alpha(m-1),1,N_b)$. How was the value of $\alpha=0.42$ selected? More specifically, what was the definition of "best fit" that was used? For $\alpha=0.42$, how do the plots of the exact $g(n,m,N_b)$ vs. n and m compare to the empirical approximation using $\alpha=0.42$?

6] In the section "Equilibrium behavior is predicted by an analytical model", l_{eff} is defined to be the effective length of a loop that contributes to the constant loop entropy term. In the section "The model is validated by comparing to exact computational enumeration ...", the paper claims that the parameter l_{eff} is fit once for the short linker and the long linker regime. But in the methods section, it is described that l_{eff} depends on the fraction of stickers bonded, which in turn depends on the number of stickers n and the binding strength F_b . According to this description, the parameter that is really fit is $l_{\text{eff}}^{\text{fit}}$ in equation 9. This is a bit confusing as they are contradictory statements. It might be worth clarifying this distinction between these two things in the main text.

7] Could physical mechanisms underlying the assumption of constant entropy of loop formation be provided?

8] Is the assumption underlying equation (9) and the fact that $l_{\text{eff}}^{\text{fit}}$ is a constant valid across a large range of n and F_b ? How does an effective loop length obtained from computational evaluation of the structures with different n and F_b compare with this analytical model?

Reviewer #2 (Remarks to the Author):

The paper by Kimchi, King, and Brenner (KKB) explores aggregation of RNA sequences, with focus on low complexity ones, using theory to calculate approximately the "phases" of such systems. The topic is of considerable interest currently, and this is in their favor. The paper is very difficult to figure out, and even after reading portions of it a few times I could not really understand it well enough to assess if it is acceptable for Nature Communications. Although the issue addressed is interesting and timely, the paper does not rise to the level needed for publication in Nature Communications even though I like this study in principle. Some of the calculations reported in the SI, although somewhat technical, are interesting.

The comments, in no particular order of importance, are given below.

General: The article is really hard to read, I suspect even for the initiated readers. It is difficult to figure out from the main text the central results in the paper and the way they connect to experiments (Ref. 7). Even the terminology used is unclear (it takes a couple of takes to figure out that two CG are clumped into one), which makes it hard to figure out the physical picture, the nature of approximation etc. This could make it hard for the general readers (or even those familiar with the problems) to get the gist of it. Some of the issues could be solved using illustration of structures (mapping of the real sequence to their model (Fig 1A does not do justice because it leaves the impression that the model is at the nucleotide level) + what is a loop in a multimer etc.). Some (not all) the specific concerns are:

(a) Out of the four figures, only Fig. 3 compares their calculations to a computer experiment. The rest are theoretical predictions, which is fine except the prospect of testing them within the context of RNA sequences is not discussed at all, as far as I can tell. How can one be assured that the results in Figs 2 and 4 could be tested, at least in principle?

(b) The authors suggest that configurational entropy (even in the abstract) as being the driving force for aggregation. From Fig. 4A, for phase separation to occur, strong interactions (more negative F_b) are actually needed for "disallowing neighbor binding". For the case of "allowing neighbor binding", there is a balance between enthalpy-entropy for phase separation to happen. If entropy is dominated, the aggregates should be present when $F_b = 0$, which is clearly not the case. Could the author elaborate?

(c) What exactly is a multimer? It seems that if there are M chains (for concreteness M CAG n polymers) is the multimer constructed by concatenation the M chains? That seems to be the logical interpretation. If so, are the authors asserting or assuming that $Z_m = Z_M$ where Z_M is the partition of M interacting chain in a box of volume V leading to the number density M/V ?

(d) Related to (c), it is unclear how the concentration in μM in Fig. 3 determined? On page 7 of the SI the authors write down some equations, which follow one from another although they could do a better job of explaining Z_m' (partition function with a prime). The standard physical definition of concentration $c = M/V$ with M and V both going to infinity such that the ratio is finite. How is the physical definition of c related to what is produced on page 7 of the SI, although there is no direct link between Eq. (S12) and $c = M/V$?

(e) There is a sentence on page 3 "We enumerated the monomer and dimer partition functions computationally, and used the analytical model to extrapolate to $m = 64$ ". I did not understand (1) how the extrapolation was done? (b) What is special about $m = 64$?

(f) In the sentence following the one quoted above it is asserted that the simulations are decidedly out of equilibrium. How was this assessed?

(g) It seems that the definition of "pseudoknot" is carried over to multimers (k-mers) as well. How do the authors define "pseudoknot" in k-mers? One could imagine for larger and larger k, the contribution to the partition function maybe increasingly dominated by such "pseudoknots". The errors reported in Fig. S5 could be larger if one goes to higher k, and not stop at dimer $k=2$ of a short chain $n=7$. Should ignoring the pseudoknot affect the phase diagram presented in Fig. 3 and 4C?

(h) It seems that the authors also accounted for loop entropy even between different chains. Why? For example, Fig. S1c, why does the term TdS_{loop} (6) appear in the free energy of the trimer?

(i) The authors write "our equilibrium results may even be more relevant in vivo than in vitro", because in vivo aggregates are more fluid-like and dynamic than in vitro. Can they elaborate on that? Could the results say something about the properties of the aggregates other than the phase diagram?

(j) Another related point: why does this method, equilibrium calculations, overestimate the concentration at which phase separation occurs? Experiments by Jain & Vale showed that phase separation happens in sub-micromolar regime.

(k) How exactly is the mapping between sequence and F_b done? Mapping between sequence and stickers & spacers?

(l) Back to entropy. In the SI they argue (perfectly OK) that Eq. 6 in the main text is dominated by the Nb^* . The calculation is fine. However, Nb^* depends on F_b . Does that not mean "enthalpy" also contributes to the driving force? In what sense is the driving force purely entropic? This is not explained well.

(m) Regarding reentrant transition: One thinks about this as a transition from phase 1 to phase 2 and then phase 2 when something is tuned. If I understand it correctly (Figure 4C) the tuning parameter is F (not controlled in experiments!) the phases are low density to high density to low density. Is this correct? If so are the two low density phases the same?

Reviewer #3 (Remarks to the Author):

This paper presents a statistical model to study the formation of RNA aggregates. The model defines the partition function describing the formation of a series of m monomers. Through this approach, the authors are able reconstruct previous results of MD simulation performed in another paper.

I do like the paper overall - this is a way to make theory and experiments closer, that for me it is quite important.

The paper is well written, but it is a little bit short in the introduction of the model and the explanation. I suggest to expand and discuss those parts.

Major comments:

1 Statistical mechanics is at equilibrium,, biology is not. Where do you approximate the equilibrium assumption ? Discuss this point is crucial for the introduction of the model.

2 it is not clear how the F_b energy contribution of the stickers has been computed. Similarly for DSloop, it is not clear how you define it. Neither the ΔF effective energy of multimerization is well defined, these are critical concept of you model IO would like to have well defined in the main text. In general I would like to have a better explanation of the model and careful physical explanation of the parameters.

3 Definition of phase transition. I do not see a well defined parameter. The authors define aggregation in terms of : 'Aggregation occurs in the parameter regime where the concentration of multimers comprised of m strands, c_m , increases with m ', why it is the phase transition range of parameters?

4 in the text operas only Z_m , but in the figure Z has multiple indexes (fig 2b for instance) what do they represent?

5 The model could be used to compute the aggregation propensity of a given transcript. Could you show and analyse some real transcript? It is possible to match your prediction with some experimental data

6 the definition of links and stickers should be more precise: for instance CGACGACGACGACGA what defines CG as the sticker and A as the linker, instead of GA the sticker and C the linker? Do I miss something?

7 in the results section the introduction of the parameters like c_m is a bit sloppy....it is not defined and only introduced.

Reviewer #1

To clarify our responses, we put sections reprinted from the manuscript in **red**, while new discussions added to the manuscript are in **blue**. Reviewer comments are in *italics*.

Summary

Motivated by prior experimental and simulation studies that show that repetitive RNA sequences can exhibit gelation, the authors of this study have developed an analytical framework to understand the effect of linkers, interaction strength, and number of stickers on the aggregation behavior of associative polymers. The authors explicitly make a distinction between intermolecular and intramolecular bonds between the polymer molecules and take into account the corresponding energy and entropy effects. While this concept has been explored before, the authors adopt a new approach where they analytically compute partition functions of molecular clusters of each size, which can then be used to compute the concentration of clusters of each size. They define aggregation as the situation where the concentration increases with cluster size, where large clusters have the highest concentration. This is a nicer criterion than the Flory-Stockmayer criterion commonly used in associative polymer theory to define the aggregation threshold. Using this criterion, they define threshold concentrations for aggregation and study the impact of varying the interaction strength, number of stickers, and linker length. The paper presents a phase diagram and has some interesting insights. Most interestingly, the authors discover a re-entrant behavior in the aggregation behavior, where aggregation initially increases and then decreases upon increasing the binding strength of the stickers. Their model reveals the underlying physical mechanism. The authors' analytical framework can be helpful for understanding mechanisms, and it obviates the need for very expensive MD simulations. Therefore, it is suitable for publication in Nature Comms. However, a number of points need to be addressed before publication can be recommended

Specific comments:

1] In 2004, Dobryin was the first to build a model for associative polymer aggregation moving beyond the mean-field approaches. Competition between intra- and inter-molecular bonds was also explicitly accounted for in this paper. However, Dobryin mainly looked at the weak/intermediate binding regime and showed that taking into account this competition leads to a change in the boundary of the aggregation line in the temperature-volume fraction space. It is important to cite this paper.

We thank the reviewer for the suggestion and indeed agree this paper is entirely relevant. We have amended the text to highlight this paper:

While intra-chain interactions are typically ignored, exceptions do exist. These include Dobrynin's 2004 study extending the Flory-Stockmayer approach to include intra-chain associations [23], and a recent publication by Weiner *et al.* which found that self-bonds play a crucial role in determining phase behavior in a lattice system with heterotypic binding motifs of varying lengths [24].

2] The discussion of G_{assoc} and F , both in the main text and S2 is unclear. Eq. 6 seems correct if F is properly calculated. Why do we need the extra factors involving F and G_{assoc} ? This point should be more clearly described.

We thank the reviewer for the comment. First, we have changed Eq 6 (now Eq 7) to match the corresponding main-text equation so that there is greater clarity. We have also included a new paragraph explaining why the term G_{assoc} is needed:

In order to fit experimental data on the prevalence of multiple nucleic acid strands binding to one another *in vitro*, nucleic acid models include a free energy penalty for multimerization. This leads to the term $(m-1) \Delta F$ in (2). This penalty is motivated by the enthalpic and entropic costs of nucleic acids binding, including ion effects and the translational and orientational entropies lost upon association. This penalty scales linearly with the number of strands in a multimer, such that each additional strand added to a multimer carries the same penalty.

ΔF plays the same role as G_{assoc} , and therefore has the same origin.

3] Further justification is needed for the use of the maximum term approximation, which implies that the typical value thus obtained is dominant.

We thank the reviewer for the suggestion and have created a figure (Supplementary Figure S3) that probes the accuracy of the saddlepoint approximation. We show the full sum as well as the ratio of the approximated sum to the full sum. We consider four versions of the saddlepoint approximation: the first-order and second-order-corrected approximations in both the strong and intermediate regimes. We find that the strong approximation (last term) and intermediate approximation (N_b^* term) provide good agreement with the full sum in their respective regimes, with the second-order-corrected approximations improving the accuracy.

The full sum calculating Z_1 (with the Stirling approximation made for the factorials) is plotted alongside four versions of the saddlepoint approximation: the first-order (green and yellow, dashed) and second-order-corrected (pink and magenta, dotted)

approximations in both the strong and intermediate regimes. Panel A shows the full partition function; panel B shows the ratio of the estimated partition function to the full sum. A dashed vertical line plots the expected crossover point between the strong and intermediate regimes. Indeed, the strong approximation (last term) and intermediate approximation (N_b^* term) provide good agreement with the full sum in their respective regimes, with the second-order-corrected approximations improving the accuracy.

4] *In the discussion, the authors state the results from Ankur Jain and Ron Vale's paper that in vivo RNA aggregates are more fluid-like and this is likely a consequence of active processes. They use this to claim that "Thus, our equilibrium results may be even more relevant in vivo than in vitro". The authors are inherently assuming that active processes that contribute to maintaining fluidity affect only the dynamics. In principle, these active processes can lead to non-equilibrium steady states that may be different from the equilibrium state.*

The reviewer makes a good point. While we had intended this sentence only as a speculation of one possible outcome, we have removed this line, and instead the text now reads:

In vivo RNA aggregates are even more fluid-like and dynamic than in vitro aggregates, for reasons that remain largely unclear but appear to be the result of active enzymes in the cell. Future work may consider how such active processes affect the aggregation properties, and the connection between in vivo non-equilibrium steady states and the equilibrium steady state discussed here.

5] *When heuristically computing $g(n,m,N_b)$ for multimers in the case of short linkers where neighbor binding is not allowed, $g(n,m,N_b)$ was approximated as $g(mn+\alpha(m-1),1,N_b)$. How was the value of $\alpha=0.42$ selected? More specifically, what was the definition of "best fit" that was used? For $\alpha=0.42$, how do the plots of the exact $g(n,m,N_b)$ vs. n and m compare to the empirical approximation using $\alpha=0.42$?*

We thank the reviewer for the insightful questions and have included a new supplementary figure (Fig. S1) to show the result of the approximation. We show the true result for g disallowing neighbor bonds for a range of n 's and m 's (limited by ensuring a reasonable computation time). The true result is calculated using Eqn. S8, which does not include the connected multimer correction; considering only connected multimers makes the calculation only feasible for even smaller values of n and m than

those used. We then also plot the results for $\alpha=0, 0.42, \text{ and } 1$, and show that $\alpha=0.42$ does a reasonable job of fitting the true result especially for large values of N_b , which are those most frequently considered in the manuscript.

For very weak interactions, such that lower values of N_b are needed, $\alpha=0.42$ is not the best fit; indeed, this value was chosen heuristically, since a single value of α cannot fit all possible regimes. For certain definitions of best-fit (e.g. ensuring that for $m=2$ $g(n, m, N_b)$ is well-fit for a large range of n) the best-fit value is indeed $\alpha=0.42$; but for example the best-fit value for $m=3$ under the same conditions is slightly smaller (~ 0.40), and it is computationally infeasible to generate enough data points for $m=4$ to test the results (though fitting only for small n suggests 0.42 would be a good fit for $m=4$ as well). Because the full sum is so computationally intensive to compute, we use a heuristic value for α which we show is good enough to reasonably fit the data. Our claim is merely that using this value of α provides a “good-enough” fit for the data, and accuracy would likely be improved by using the full sum (Eqn. S8). Further work may find a better formula to fit this full sum, but it is outside the scope of this work.

The exact calculation of $g_d(n, m, N_b)$ (S8) is shown alongside the heuristic approximations $g_d(nm + \alpha(m-1), 1, N_b)$ for $\alpha=0, 0.42$ and 1 . The range of n and m chosen was limited by the computation of the full sum. $\alpha=0.42$ shows good agreement with the full sum over this range especially for large values of N_b , which are most relevant in the strong interaction regime. A better

fit may be found by modulating α as a function of F_b , or perhaps with a different model.

We have also amended the text (both in the main text and the supplement) to clarify:

The value of $\alpha=0.42$ used is a heuristic estimate that is an especially good fit to the strong interaction regime, and other approximations may improve it.

6] In the section “Equilibrium behavior is predicted by an analytical model”, l_{eff} is defined to be the effective length of a loop that contributes to the constant loop entropy term. In the section “The model is validated by comparing to exact computational enumeration ...”, the paper claims that the parameter l_{eff} is fit once for the short linker and the long linker regime. But in the methods section, it is described that l_{eff} depends on the fraction of stickers bonded, which in turn depends on the number of stickers n and the binding strength F_b . According to this description, the parameter that is really fit is l_{eff}^{fit} in equation 9. This is a bit confusing as they are contradictory statements. It might be worth clarifying this distinction between these two things in the main text.

We thank the reviewer for the comment. Indeed, we seem to have conflated l_{eff} and l_{eff}^{fit} in the text. The text has now been corrected: the parameter fit is l_{eff}^{fit} , which enters the model through its dependence on l_{eff} . The text now reads:

A single fitting parameter, corresponding to the normalized effective loop length, l_{eff}^{fit} , is used.

7] Could physical mechanisms underlying the assumption of constant entropy of loop formation be provided?

We thank the reviewer for the insightful question. We have added this discussion to the Methods section, reprinted below:

This approximation is reasonable because of two factors. First, because of the logarithmic dependence of ΔS_{loop} on loop length (Eqn. 5), moderate heterogeneities in loop length lead to only small differences in ΔS_{loop} . Second, because the typical number of loops in a multimer scales linearly with the size of the multimer (see Supplementary Section S3), we expect similar levels of heterogeneity in loop length independent of the size of the multimer. This approximation is expected to break down for very large n and weak binding ($F_b > 0$), in which

case the few loops that typically form will likely have a broad distribution of lengths; this regime is not considered here.

8] Is the assumption underlying equation (9) and the fact that $l_{\text{eff}}^{\text{fit}}$ is a constant valid across a large range of n and F_b ? How does an effective loop length obtained from computational evaluation of the structures with different n and F_b compare with this analytical model?

We thank the reviewer for the question. Indeed, this assumption is valid across a large range of n , m , and F_b . In Fig. 2, we show that in the strong interaction regime, the assumption is valid across a range of n and m . In Fig. S5 (previously S3), we show that even when using the value of $l_{\text{eff}}^{\text{fit}}$ from the strong interaction regime, the same value shows good agreement across a range of n and m .

We have added text to clarify that despite these results, this agreement is not expected to last indefinitely:

This approximation is expected to break down for very large n and weak binding ($F_b > 0$), in which case the few loops that typically form will likely have a broad distribution of lengths; this regime is not considered here.

The final question the reviewer asks is an important one. Indeed, while we quoted the values fitted for $l_{\text{eff}}^{\text{fit}}$, we did not provide context for them. In fact, the best-fit values are very close to what we would naively expect from looking at the structures in the strong interacting regime (for which $l_{\text{eff}} \sim l_{\text{eff}}^{\text{fit}}$). We have added the following discussion to the Methods section:

We fit $l_{\text{eff}}^{\text{fit}}$ to the strong binding regime (Fig. 2) for which $l_{\text{eff}} \approx l_{\text{eff}}^{\text{fit}}$. We find intuitively reasonable values for $l_{\text{eff}}^{\text{fit}}$. When using $l=1$ (disallowing neighbor binding), we find $l_{\text{eff}}^{\text{fit}} = 4.3$ nucleotides. This value is in between the length of an internal loop formed by two individual linkers (4 nucleotides) and the length of a hairpin loop formed by two linkers and a sticker (5 nucleotides). When using $l=4$ (allowing neighbor binding), we find $l_{\text{eff}}^{\text{fit}} = 7$ nucleotides. This value is also in between the length of an internal loop formed by two individual linkers (10 nucleotides) and the length of a hairpin loop formed by a single linker (5 nucleotides).

Reviewer #2

To clarify our responses, we put sections reprinted from the manuscript in **red**, while new discussions added to the manuscript are in **blue**. Reviewer comments are in *italics*.

Summary

The paper by Kimchi, King, and Brenner (KKB) explores aggregation of RNA sequences, with focus on low complexity ones, using theory to calculate approximately the “phases” of such systems. The topic is of considerable interest currently, and this is in their favor. The paper is very difficult to figure out, and even after reading portions of it a few times I could not really understand it well enough to assess if it is acceptable for Nature Communications. Although the issue addressed is interesting and timely, the paper does not rise to the level needed for publication in Nature Communications even though I like this study in principle. Some of the calculations reported in the SI, although somewhat technical, are interesting.

The comments, in no particular order of importance, are given below.

General comment:

General: The article is really hard to read, I suspect even for the initiated readers. It is difficult to figure out from the main text the central results in the paper and the way they connect to experiments (Ref. 7). Even the terminology used is unclear (it takes a couple of takes to figure out that two CG are clumped into one), which makes it hard to figure out the physical picture, the nature of approximation etc. This could make it hard for the general readers (or even those familiar with the problems) to get the gist of it. Some of the issues could be solved using illustration of structures (mapping of the real sequence to their model (Fig 1A does not do justice because it leaves the impression that the model is at the nucleotide level) + what is a loop in a multimer etc.).

We thank the reviewer for their careful reading of the manuscript. We have implemented many changes to make the paper clearer and easier to understand. In addition to many modifications of the text (described in more detail below in response to individual concerns) we have a new version of Fig. 1 that we believe addresses the points that were causing confusion. The first panel of Fig. 1A shows two different sequences to demonstrate how groups of nucleotides are grouped into stickers and linkers, and how the free energy is calculated as a function of stickers. We show an example of both short and long linkers, and show that depending on the sequence, A's or C's can be part of either stickers or linkers. In the third panel of Fig. 1A, we show the structure enumerate procedure at the level of the sticker-spacer model rather than at the

nucleotide level. In order to clarify this issue of loops in a multimer, we also spell out how many stems and loops each of the example structures we provide has. As can be seen from the dimer structure, both intra- and inter-molecular stems can lead to loops with the same physical properties. The reader may notice, for example, that there is an internal loop in the top-left of the dimer structure flanked by two intramolecular stems, which is indistinguishable in structure from the four internal loops flanked by intermolecular stems at the base of the structure. There is no difference between loops in a multimer and loops in a monomer.

Specific comments:

Some (not all) the specific concerns are:

(a) Out of the four figures, only Fig. 3 compares their calculations to a computer experiment. The rest are theoretical predictions, which is fine except the prospect of testing them within the context of RNA sequences is not discussed at all, as far as I can tell. How can one be assured that the results in Figs 2 and 4 could be tested, at least in principle?

We thank the reviewer for the insightful question. Our model predictions, specifically Figs. 4 & 5, could be tested experimentally by measuring the concentration at which aggregates start forming for sequences with different stickers and different linker lengths. These measurements are not too difficult to perform (see e.g. “Divalent cations can control a switch-like behavior in heterotypic and homotypic RNA coacervates” (2019) which computes a similar quantity as a function of ion concentration instead of RNA concentration). Fig. 2 depicts calculations internal to the model, and could not easily be tested experimentally; Figs 4 & 5 depict true model predictions.

We would indeed love to see a further experimental comparison with our model. Our results are in qualitative agreement with the Jain & Vale experiments showing that larger values of n lead to a higher degree of aggregation. Beyond this comparison, the closest we are able to come in our study to compare to experimental results is our comparison to molecular dynamics (MD) simulations (Fig. 3). We showed that with no parameter tuning whatsoever, our model is able to quantitatively replicate the results of these simulations. We have added the following to our discussion section:

These effects, and the predicted phase diagram as a whole (Fig. 5B) could be at least qualitatively tested experimentally by replicating the Jain & Vale experiments for multiple sequences with different sticker strengths and linker lengths, and measuring the change

in the concentration needed to form aggregates for the different conditions. The available published data is in good agreement with our predictions, in that larger values of ϕ show a greater propensity for aggregation in both experiments and our model predictions.

(b) The authors suggest that configurational entropy (even in the abstract) as being the driving force for aggregation. From Fig. 4A, for phase separation to occur, strong interactions (more negative F_b) are actually needed for "disallowing neighbor binding". For the case of "allowing neighbor binding", there is a balance between enthalpy-entropy for phase separation to happen. If entropy is dominated, the aggregates should be present when $F_b = 0$, which is clearly not the case. Could the author elaborate?

We thank the reviewer for the insightful question. A major point we seek to drive home is that in contrast to the prevailing view in the field which considers aggregation to occur when more than some critical number of bonds is satisfied (e.g. Semenov & Rubenstein), a more careful analysis reveals that having more bonds be satisfied does not necessarily lead to more aggregation.

To understand why configurational entropy is the driving force for aggregation, consider the three terms in the free energy (Eqn. 2; previously, Eqn. 1). The three terms are ΔF , F , and g . The first is a multimerization cost; it penalizes large multimers, and therefore not only doesn't drive aggregation, but suppresses aggregates. The second is the enthalpy. We show that this term is equally well satisfied by a group of dimers that have all their stickers bound to one another as by a large aggregate with all bonds satisfied. Therefore, this cannot be the driving force for aggregation. If these two terms were the only terms in the free energy, we would not see any aggregates. The second term does nothing to distinguish between aggregates and smaller clusters, while the first term suppresses aggregates. Thus, aggregation as a whole would never be observed in the system.

The reason we do indeed see aggregates is because of the third term, the configurational entropy. Of the three terms in the free energy, only the configurational entropy can bias the system towards forming aggregates. This explanation is given in the Discussion section.

In this system, aggregation is not necessarily predicted as the regime where the most possible bonds are satisfied, as bonds can be satisfied by intramolecular as well as by intermolecular contacts. Instead, aggregation is predicted by the relative stability of the

aggregate compared to smaller multimers. The stability of each structure is a function of three terms: 1) the number of stickers bound (each contributes F to the free energy); 2) the number of strands in the structure (each contributes $\mu + \Delta F$); and 3) the configurational entropy of the structure. This last term contributes $-\log(g)/\beta$ to the free energy, where g is the number of ways to satisfy the given number of bonds with the given number of strands in the structure.

This last term is the driving force for aggregation in this system. Aggregates are no more stable than dimers in terms of the first term, the possible number of stickers bound (both are able to satisfy all stickers). Aggregates are further penalized by the second term, the multimerization cost. If these two terms were the only terms in the free energy, we would not see any aggregates. It is the third term, the configurational entropy, that drives aggregation. Larger multimers are able to satisfy their bonds in many more configurations than a corresponding collection of smaller multimers, leading to an enormous entropic benefit in forming aggregates. This has been described as a competition between configurational and translational entropies in other contexts. In our system, the benefit due to g peaks when most, but not all, stickers are satisfied (Fig. S6).

The way the equilibrium distribution of structures is found by the system can be thought of as a two-step process. First, the system decides what fraction of stickers should be bound. This is determined by the enthalpic benefit of forming stickers. Then, the system decides how to partition the structures while forming this number of stickers. (Of course, we do not mean this occurs literally – this is simply a way to intuitively understand the behavior of the system).

As seen in the new Supplementary Fig. S6, the maximum of the configurational entropy benefit of forming an aggregate occurs when most stickers are satisfied. In other words, the system is most likely to partition itself into an aggregate cluster not when the enthalpy is zero, but when the enthalpy is low enough that most stickers are bound. This is particularly true in the regime disallowing neighbor binding, where the maximum of the entropic benefit the system gains from forming an aggregate occurs when nearly all stickers are satisfied.

This is indeed what our model predicts. The figure to look at is Fig. 5 (Fig. 4 which the reviewer mentions has F_b on the x-axis, which mostly serves to counteract loop entropy. Fig. 5 has the true sticker strength, which is a combination of enthalpy and loop entropy, F , on the x-axis). Indeed, for the case of allowing neighbor binding, aggregation probability is maximized for slightly negative F , for which most, but far from all, stickers are typically bound. For the case of disallowing neighbor binding, aggregation

probability is maximized for more negative F ; in this regime, nearly all stickers are typically bound.

The aggregation behavior of the system as a whole (Fig. 5) is thus well-predicted by considering only the configurational entropy component (Supplementary Fig. S6).

The figure caption for this new figure reads:

Configurational entropy drives aggregation. The ratio of the configurational entropy of an m -mer to the configurational entropy of m monomers with the same total number of stickers bound, N_b , is shown as a function of N_b normalized by the maximum value it can take, $N_b^{\text{max}} = n m / 2$. This ratio can serve as a proxy for the propensity of the system to aggregate based on configurational entropy considerations alone. The typical number of bonds satisfied in the system, or where on the x-axis the system will typically lie, is determined by the sticker strength $-\beta F$. The configurational entropy ratio is maximized when sticker strength $-\beta F$ is large enough such that most, but not all, stickers are bound. For the case of disallowing neighbor binding, aggregation is most likely when nearly all stickers are bound. That the maximum of the ratio plotted does not occur at the maximum value of N_b leads to the reentrant phase transition explored in this work.

We have also appended the following to the main-text discussion of Fig. 5:

The behavior shown in Fig. 5 is in agreement with what we would expect from configurational entropy concerns alone (Fig. S6). That the propensity of the system to aggregate occurs at more negative values of $-\beta F$, and is more pronounced, for the case of disallowing neighbor binding than for the case of allowing neighbor binding, is predicted by the different forms of the configurational entropy in these two regimes.

Similarly, larger values of $\beta\Delta F$ increase the propensity of the system to aggregate because of their effect on configurational entropy, rather than any enthalpic considerations (Fig. S6)

(c) What exactly is a multimer? It seems that if there are M chains (for concreteness M CAG n polymers) is the multimer constructed by concatenation the M chains? That seems to be the logical interpretation. If so, are the authors asserting or assuming that $Z_m = ZM$ where ZM is the partition of M interacting chain in a box of volume V leading to the number density M/V ?

We thank the reviewer for the question. We define a multimer based on which strands are bound to one another through sticker-sticker bonds. In addition to the revised Fig. 1A, which we hope clarifies the issue somewhat (especially the picture of the dimer in panel 3), we have added the following paragraph to the model description in the main text to clarify:

In this work, we are concerned with the behavior resulting from such sequences interacting with one another. Two stickers that bind to one another may be on the same strand or on two different strands. Moreover, many strands can be connected to one another through a chain of such bonds. We call a group of m strands connected through a series of intermolecular bonds a multimer of size m , or an m -mer. There are many ways a multimer of size m can form: any combination of bonds that occur either intra- or inter-molecularly within a group of m strands, such that each strand is reachable from every other by following a series of intermolecular bonds, is an m -mer.

(d) Related to (c), it is unclear how the concentration in M in Fig. 3 determined? On page 7 of the SI the authors write down some equations, which follow one from another although they could do a better job of explaining Z_m' (partition function with a prime). The standard physical definition of concentration $c = M/V$ with M and V both going to infinity such that the ratio is finite. How is the physical definition of c related to what is produced on page 7 of the SI, although there is no direct link between Eq. (S12) and $c = M/V$?

We thank the reviewer for the question. Supplementary section S2 explains the units chosen for the concentration, and why it is natural for those units to be $\exp(-\beta\Delta F)$. We have updated the text of the section to define the partition function with a prime more clearly:

$$Z_m' \equiv Z_m e^{\beta(m-1) \Delta G_{\text{assoc}}}$$

The reviewer's definition of concentration is the same as our own. We have added this explicitly to the main text:

We consider a system of M strands present in a container of volume V , such that their concentration is $c^{\text{tot}}=M/V$. We take the thermodynamic limit of M and V going towards infinity with their ratio staying constant.

We have also added the definition of the concentration of multimers to the main text, alongside a new equation (now equation 1).

We define c_m as the concentration of multimers of size m , such that $c^{\text{tot}} = \sum_{m=1}^{\infty} m c_m$.

(e) There is a sentence on page 3 "We enumerated the monomer and dimer partition functions computationally, and used the analytical model to extrapolate to $m = 64$ ". I did not understand (1) how the extrapolation was done? (b) What is special about $m = 64$?

We thank the reviewer for the comment. This number was chosen to match the MD simulations we compare against. Previously in the paragraph, we had written

Those simulations examined 64 CAG-repeat RNA strands with varying numbers of repeats and concentrations.

We have amended the text so that this number is clarified again when we describe our calculation. The text now reads:

We enumerated the monomer and dimer partition functions computationally, and used the analytical model to extrapolate up to $m=64$, the number of strands used in the MD simulations. The extrapolation was performed by fitting the single parameter to our computational results for $m=1$, and using (S38) and (S48) to obtain the results for $m>2$.

(f) In the sentence following the one quoted above it is asserted that the simulations are decidedly out of equilibrium. How was this assessed?

We thank the reviewer for the question. Several of the figures in the simulation paper show time traces of various quantities that are clearly continuing to change in a directional manner after the 10^8 timesteps for which the simulations were performed. See for example Figs. 1a, 3a, S3, and S4. Furthermore, the authors write in their paper that they believe “our simulations could be near the onset of coarsening into a rigid gel-like state”, in contrast to an equilibrium state.

(g) It seems that the definition of "pseudoknot" is carried over to multimers (k-mers) as well. How do the authors define "pseudoknot" in k-mers? One could imagine for larger and larger k, the contribution to the partition function maybe increasingly dominated by such "pseudoknots". The errors reported in Fig. S5 could be larger if one goes to higher k, and not stop at dimer k=2 of a short chain n=7. Should ignoring the pseudoknot affect the phase diagram presented in Fig. 3 and 4C?

We thank the reviewer for the detailed question, and indeed the same issues concerned us.

We'll first start with the definition of pseudoknots in m-mers. A simple way to define pseudoknots in a monomer is to draw the nucleotides in a circle (such that the first nucleotide is near the last), and then to draw lines connecting nucleotides base-paired to one another. Lines that cross define a pseudoknot. The same thing can be done for multimers, with concatenating the sequences before drawing them in a circle. The entropic cost of forming pseudoknots depends on the particular pseudoknot formed. (Many) more details can be found in our 2019 paper “A polymer physics framework for the entropy of arbitrary pseudoknots” as well as other publications (cited there).

Ultimately, the inclusion of pseudoknots may indeed affect the results, but we believe the excellent agreement between our model's predictions and MD simulation results—which by their nature do nothing to exclude pseudoknots and indeed contain many intermolecular pseudoknots—demonstrates that considering pseudoknots is not key to understanding the process we study here.

We have amended the text to clarify that the no-pseudoknot assumption is just that (an assumption):

We make the approximation that the contribution of pseudoknots to the partition function is negligible due to their high entropic cost.

We have also removed the following text from the supplement, claiming that the error introduced by our no-pseudoknot assumption is insignificant:

~~The effects of disallowing pseudoknots therefore do not appear particularly significant for our purposes, and we do not consider pseudoknots further.~~

Our manuscript therefore does not now claim that including pseudoknots would have a negligible effect on the results. While we do indeed believe this to be the case (as discussed above), it is very difficult to demonstrate this conclusively. We would be very interested in the results of future studies which may probe just how pseudoknots affect the results we discuss here. However, for the purposes of this study, it is remarkably difficult to enumerate the entire structure landscape of larger multimers including pseudoknots to probe the effect pseudoknots have on the landscape.

We used LandscapeFold to consider the results shown in the supplementary section on pseudoknots. LandscapeFold has the benefit of explicitly enumerating all structures that can form, such that we can explicitly see how pseudoknots affect the landscape. However, in doing so, the computation time grows exponentially with n . As shown in the previous version of Fig. S4, (now Fig. S7) the “short chain” cited by the reviewer for $n=7$, $m=2$ had over 10^6 enumerated structures, and $n=8$ simply could not be computed in reasonable time & memory. LandscapeFold cannot consider trimers or higher-order multimers.

Other algorithms exist to consider pseudoknots in RNA structure, but not (to our knowledge) for multiple strands with the calculation we need here. For example, NuPack can consider pseudoknots, but only for a single strand; a new algorithm VfoldMCPX can consider pseudoknots across multiple strands, but does not return the partition function (only the top few structures); and so on. Even those algorithms that can perform the partition function calculation for monomer pseudoknots only consider a restricted class of pseudoknots: the problem of predicting the minimum free energy structure of a sequence including pseudoknots is NP-complete.

(h) It seems that the authors also accounted for loop entropy even between different chains. Why? For example, Fig. S1c, why does the term TdS_{loop} (6) appear in the free energy of the trimer?

To clarify this point, we have amended Fig. 1 as previously discussed. In the third panel of Fig. 1A, we spell out how many stems and loops each of the example structures we provide has. The dimer structure demonstrates that both intra- and inter-molecular

stems can lead to loops with the same physical properties. For example, there is an internal loop in the top-left of the dimer structure flanked by two intramolecular stems, which is indistinguishable in structure from the four internal loops flanked by intermolecular stems at the base of the structure. There is no difference between loops in a multimer and loops in a monomer. This phenomenon is well-known, see e.g. “Thermodynamic analysis of interacting nucleic acid strands” (2007) (Ref. 32 in our manuscript).

(i) The authors write "our equilibrium results may even be more relevant in vivo than in vitro", because in vivo aggregates are more fluid-like and dynamic than in vitro. Can they elaborate on that? Could the results say something about the properties of the aggregates other than the phase diagram?

The reviewer asks excellent questions. In response to a comment from a different reviewer, we have removed this line, and instead the text now reads (blue is new added text):

In vivo RNA aggregates are even more fluid-like and dynamic than in vitro aggregates, for reasons that remain largely unclear but appear to be the result of active enzymes in the cell. Future work may consider how such active processes affect the aggregation properties, and the connection between in vivo non-equilibrium steady states and the equilibrium steady state discussed here.

(j) Another related point: why does this method, equilibrium calculations, overestimate the concentration at which phase separation occurs? Experiments by Jain & Vale showed that phase separation happens in sub-micromolar regime.

We thank the reviewer for the question. It does not appear that these equilibrium calculations overestimate the concentration at which phase separation occurs; see for example Fig. 3 which predicts this concentration with quite high accuracy. Instead, the reason for the discrepancy between our results and the Jain & Vale experiments appears to be due to Magnesium. As we write:

A limitation of our model's physiological applicability is that we did not explicitly consider magnesium. Magnesium can act as a bridge between negatively-charged RNA molecules such that even in the absence of base pairing, Mg-RNA mixtures can form aggregates. Experimental results thus rely on magnesium aiding the aggregation process. However, the MD simulations we compare to here do not explicitly consider

Magnesium and the high concentrations required for the system to aggregate (e.g. Fig. 3) are the result.

We further write in the Fig. 4 caption:

The high concentration is a result of the lack of Mg^{+2} considered explicitly in the model.

(k) How exactly is the mapping between sequence and Fb done? Mapping between sequence and stickers & spacers?

We thank the reviewer for the questions, and have amended Fig. 1A to clarify. In the first panel, we give two example sequences, and show how the three main parameters in the sticker-spacer model (n , l , and F_b) are calculated for each sequence.

n is the number of repeats in a strand. l is the number of nucleotides in the linker. F_b is calculated through the classic nearest-neighbor model for RNA or DNA base-pairing interactions. In short, each stem is broken up into a sequence of overlapping two-base-pair-long windows, each of which has a free energy parameterized by experiments. The free energy of the stem as a whole is well-approximated by the sum of the free energies of its component “nearest-neighbor” base pairs. This procedure is described in many references; we have cited a few, as discussed below.

The results of our paper are entirely invariant to how exactly the mapping between sequence and F_b is achieved. The nearest-neighbor model is simply the simplest and most common way of performing this mapping with reasonably good accuracy.

In addition to modifying the figure, we have modified both the main text and the Methods section with the following additions:

We consider a nucleic acid sequence comprised of n identical stickers (Fig. 1A). The stickers are separated by $n-1$ equally-spaced linkers that do not interact with the stickers. Each linker consists of l nucleotides.

$[F_b]$ is calculated using the classic nearest-neighbor model for RNA or DNA base-pairing interactions \cite{Turner2009, SantaLucia2004}. The linkers, each of which is of length l , are inert.

(l) Back to entropy. In the SI they argue (perfectly OK) that Eq. 6 in the main text is dominated by the Nb^ . The calculation is fine. However, Nb^* depends on Fb . Does that not mean “enthalpy” also contributes to the driving force? In what sense is the driving force purely entropic? This is not explained well.*

See our response to comment (b) and the new Supplementary Figure S6.

(m) Regarding reentrant transition: One thinks about this as a transition from phase 1 to phase 2 and then phase 2 when something is tuned. If I understand it correctly (Figure 4C) the tuning parameter is F (not controlled in experiments!) the phases are low density to high density to low density. Is this correct? If so are the two low density phases the same?

We thank the reviewer for the insightful comments and questions. First of all, the tuning parameter is indeed βF (the dimensionless strength of each sticker). This parameter is directly controlled in experiments. F can be varied as a function of the sticker sequence, as shown in Fig. 4 (top axis); moreover, F can be changed for a single sequence by modulating ionic concentrations (see e.g. Koehler & Peyret, 2005) and by modulating temperature. These latter two will also vary $\beta \Delta F$, the multimerization cost, and so sequence variation is the most straightforward way to vary F directly.

Indeed, the reentrant transition is from a low density phase to high density and back. The two low density phases are very different from one another: in the strong binding regime (very negative βF), all or nearly all bonds are satisfied in a typical molecule; in the weak binding regime ($\beta F \sim 0$), very few bonds are satisfied in a typical molecule.

We have created a new version of Fig. 4, splitting the previous figure into two new figures, and adding a pictorial representation of this reentrant transition—and the difference between the two low-density phases—to the new version of Fig. 4.

a Allowing neighbor binding

b Disallowing neighbor binding

The new figure caption reads, in part:

Agregates are most likely to form for intermediate sticker strengths, since very strong stickers lead to stable monomers (red) or dimers (dimers, trimers, and tetramers comprise the orange curve). Although aggregates are suppressed in both strong (green background; left) and weak (gray background; right) binding regimes, the molecular structures of monomers and dimers in these regimes are quite different: in the former, all or nearly all bonds are satisfied in a typical molecule, while very few bonds are typically satisfied in the latter regime. For this reason, disallowing neighbor binding (i.e. the short linker case) leads to a large concentration of dimers, and few monomers, predicted in the strong binding regime. When allowing neighbor binding (i.e. the long linker case), monomers are also able to satisfy all bonds and are thus present at high concentrations in the strong binding regime.

Reviewer #3

To clarify our responses, we put sections reprinted from the manuscript in red, while new discussions added to the manuscript are in blue. Reviewer comments are in *italics*.

Summary

This paper presents a statistical model to study the formation of RNA aggregates. The model defines the partition function describing the formation of a series of m monomers. Through this approach, the authors are able to reconstruct previous results of MD simulation performed in another paper.

I do like the paper overall - this is a way to make theory and experiments closer, that for me it is quite important.

The paper is well written, but it is a little bit short in the introduction of the model and the explanation. I suggest to expand and discuss those parts.

We thank the reviewer for the suggestion, and have indeed expanded the introduction of the model and its explanation. We have a new version of Fig. 1 which includes in panel A a description of how the main parameters of the model are derived from the sequence. Furthermore, we have added four paragraphs and an equation to the main text of the results section to clarify the model:

In this work, we are concerned with the behavior resulting from such sequences interacting with one another. Two stickers that bind to one another may be on the same strand or on two different strands. Moreover, many strands can be connected to one another through a chain of such bonds. We call a group of m strands connected through a series of intermolecular bonds a multimer of size m , or an m -mer. There are many ways a multimer of size m can form: any combination of bonds that occur either intra- or inter-molecularly within a group of m strands, such that each strand is reachable from every other by following a series of intermolecular bonds, is an m -mer.

We consider a system of M strands present in a container of volume V , such that their concentration is $c^{\text{tot}} = M/V$. We take the thermodynamic limit of M and V going towards infinity with their ratio staying constant. We seek to predict how frequently multimers comprised of m strands form in this system, and how this frequency changes with m . We define c_m as the concentration of multimers of size m , such that $c^{\text{tot}} = \sum_{m=1}^{\infty} m c_m$.

There are two possible regimes for the system: For large m , c_m either decreases or increases with m (Fig. 1A). In the former case, the system is in the dilute phase, with only small multimers typically forming. In contrast, if c_m increases with m , large aggregates of the order of the system size dominate the landscape. The aggregation transition is defined as the crossover point between the regime in which very large multimers are suppressed, to that in which they are dominant.

In order to fit experimental data on the prevalence of multiple nucleic acid strands binding to one another *in vitro*, nucleic acid models include a free energy penalty for multimerization. This leads to the term $(m-1) \Delta F$ in (2). This penalty is motivated by the enthalpic and entropic costs of nucleic acids binding, including ion effects and the translational and orientational entropies lost upon association. This penalty scales linearly with the number of strands in a multimer, such that each additional strand added to a multimer carries the same penalty.

Major comments:

1) *Statistical mechanics is at equilibrium,, biology is not. Where do you approximate the equilibrium assumption ? Discuss this point is crucial for the introduction of the model.*

We thank the reviewer for the insightful question. Our equilibrium assumption comes from two factors. First, the main experiments we are motivated by (Jain & Vale, 2017) were performed *in vitro*, rather than *in vivo*, and are therefore not subject to the constant drive by biological systems to maintain life (i.e. to stay out of equilibrium).

Second, we note that despite our equilibrium approximation, we get excellent agreement with out-of-equilibrium molecular dynamics simulations (Fig. 3). This success provides an indication that the model we derive may be generally relevant even when compared to out-of-equilibrium real systems.

2) *It is not clear how the F_b energy contribution of the stickers has been computed. Similarly for DSloop, it is not clear how you define it. Neither the ΔF effective energy of multimerization is well defined, these are critical concept of you model IO would like to have well defined in the main text. In general I would like to have a better explanation of the model and careful physical explanation of the parameters.*

We thank the reviewer for the questions and have rewritten the text extensively to clarify these points. In addition, we have a new version of Fig. 1A that we believe helps with this clarification.

We'll take the terms mentioned by the reviewer one by one:

Fb:

In the first panel of the new version of Fig. 1A, we give two example sequences, and show how the three main parameters in the sticker-spacer model (n , l , and F_b) are calculated for each sequence. F_b is calculated through the classic nearest-neighbor model for RNA or DNA base-pairing interactions. In short, each stem is broken up into a sequence of overlapping two-base-pair-long windows, each of which has a free energy parameterized by experiments. The free energy of the stem as a whole is well-approximated by the sum of the free energies of its component "nearest-neighbor" base pairs. This procedure is described clearly in many references; we have cited a few, as discussed below.

The results of our paper are entirely invariant to how exactly the mapping between sequence and F_b is achieved. The nearest-neighbor model is simply the simplest and most common way of performing this mapping with reasonably good accuracy.

In addition to modifying the figure, we have supplemented the previous discussion (shown in red below) in both the main text and the Methods section with the following additions:

We consider a nucleic acid sequence comprised of n identical stickers (Fig. 1A). The stickers are separated by $n-1$ equally-spaced linkers that do not interact with the stickers. Each linker consists of l nucleotides. Stickers are self-complementary and bind through base pairing interactions, such that each sticker can be bound to at most one other. Each bonded sticker has a free energy contribution of F_b

F_b is calculated using the classic nearest-neighbor model for RNA or DNA base-pairing interactions \cite{Turner2009, SantaLucia2004}. The linkers, each of which is of length l , are inert.

DSloop:

We have supplemented the previous discussion with further explanation in the main text:

Each bonded sticker has a free energy contribution of F_b ; however, bonds that create closed loops also have an entropic cost ΔS_{loop} that depends on the loop length l_{loop} . This is because nucleotides comprising a closed loop (such as a hairpin, internal, or multi-loop) are constrained in the conformations they can adopt. A simple model treating unbound nucleotides as a polymer random walk estimates that the entropic cost of forming loops scales logarithmically with the loop length (see Methods) \cite{Jacobson1950, Kimchi2019}.

The Methods section gives the precise formula for this logarithmic dependence, which is used in the computational model:

Each closed loop of length l_{loop} leads to an entropic penalty of $\Delta S_{\text{loop}}(l_{\text{loop}})$, associated with the decrease in three-dimensional configurations of the single-stranded region of the loop compared to a free chain, given by \cite{Jacobson1950, Kimchi2019}

$$\Delta S_{\text{loop}}(l_{\text{loop}}) = k_B \left[\ln v_s + \frac{3}{2} \ln \left(\frac{3}{2} \right) \right]$$

where v_s is the volume within which two nucleotides can bind, and b is the persistence length of single-stranded regions. This equation treats the single-stranded loop as an ideal chain.

Ultimately, for our analytical model, this loop entropy is not considered explicitly, but only serves to modify F from what one would expect it to be using the nearest-neighbor model.

DeltaF:

We have included the following explanation of the origin of the multimerization penalty.

In order to fit experimental data on the prevalence of multiple nucleic acid strands binding to one another *in vitro*, nucleic acid models include a free energy penalty for multimerization. This leads to the term $(m-1) \Delta F$ in (2). This penalty is motivated by the enthalpic and entropic costs of nucleic acids binding, including

ion effects and the translational and orientational entropies lost upon association. This penalty scales linearly with the number of strands in a multimer, such that each additional strand added to a multimer carries the same penalty.

The only other parameters in the model are n and l . The former is the number of repeats per strand, and the latter is the number of nucleotides in each linker. We have amended both the main text and Fig. 1A to clarify these.

We consider a nucleic acid sequence comprised of n identical stickers (Fig. 1A). The stickers are separated by $n-1$ equally-spaced linkers that do not interact with the stickers. Each linker consists of l nucleotides.

3) Definition of phase transition. I do not see a well defined parameter. The authors define aggregation in terms of 'Aggregation occurs in the parameter regime where the concentration of multimers comprised of m strands, c_m , increases with m ', why it is the phase transition range of parameters?

We thank the reviewer for the question. Ultimately, to understand the aggregation transition, we want to understand the behavior of very large multimers (i.e. m -mers, with $m \rightarrow \infty$). These are strongly suppressed in the dilute phase, but dominate the system in the dense phase. In our formalism, we find that the concentration of multimers depends approximately exponentially on the size of the multimer. When the slope of the exponential is negative, large multimers are strongly suppressed; when the slope is positive, the opposite is true. This slope therefore determines the phase transition.

We have added the following to the main text to clarify:

There are two possible regimes for the system: For large m , c_m either decreases or increases with m (Fig. 1A). In the former case, the system is in the dilute phase, with only small multimers typically forming. In contrast, if c_m increases with m , large aggregates of the order of the system size dominate the landscape. The aggregation transition is defined as the crossover point between the regime in which very large multimers are suppressed, to that in which they are dominant.

4) In the text operas only Z_m , but in the figure Z has multiple indexes (fig 2b for instance) what do they represent?

We thank the reviewer for the comment and the opportunity to clarify. There is always only one index for the partition functions (the subscript). The superscript “index” was actually that partition function being raised to a power. We have amended Fig. 2 to include parentheses to clarify.

5) The model could be used to compute the aggregation propensity of a given transcript. Could you show and analyse some real transcript? It is possible to match your prediction with some experimental data

We thank the reviewer for the question. We would love to see a further experimental comparison with our model. One way to do so would be to replicate the Jain & Vale experiments with multiple different sequences of varying binding strengths and linker lengths. A plot of threshold concentration as a function of these two parameters could be then compared with our model predictions (Fig. 5).

Our results are in agreement with the Jain & Vale experiments showing that larger values of n lead to a higher degree of aggregation. Beyond this comparison, the closest we are able to come in our study to compare to experimental results is our comparison to molecular dynamics (MD) simulations (Fig. 3). We showed that with no parameter tuning whatsoever, our model is able to quantitatively replicate the results of these simulations.

We have added the following to our discussion section

These effects, and the predicted phase diagram as a whole (Fig. 5B) could be at least qualitatively tested experimentally by replicating the Jain & Vale experiments for multiple sequences with different sticker strengths and linker lengths, and measuring the change in the concentration needed to form aggregates for the different conditions. The available published data is in good agreement with our predictions, in that larger values of n show a greater propensity for aggregation in both experiments and our model predictions.

6) The definition of links and stickers should be more precise: for instance CGACGACGACGA what defines CG as the sticker and A as the linker, instead of GA the sticker and C the linker? Do I miss something?

We have amended Fig. 1A to clarify the origin of the stickers and linkers. As seen in the first panel, stickers are self-complementary, with the usual rules of A binding to U/T and

C binding to G. (In RNA, G and U also bind, but this doesn't come into play in this paper). The linkers are orthogonal to the stickers. To show this, we have two example sequences. In one, the stickers are comprised of GC, and A comprises the linkers; in the other, the stickers are comprised of AAUU (which is self-complementary, just as GC is) and the linkers are comprised of four C's. GA can't be the sticker simply because GA is not self-complementary. To clarify this, we write in the text

The stickers are separated by $n-1$ equally-spaced linkers that do not interact with the stickers. Each linker consists of l nucleotides. Stickers are self-complementary and bind through base pairing interactions

7) In the results section the introduction of the parameters like cm is a bit sloppy....it is not defined and only introduced.

We thank the reviewer for the comment and have modified the text to be more precise in our definition of multimers and of the concentration. We have also included a new main-text equation to further clarify this point (see below).

In this work, we are concerned with the behavior resulting from such sequences interacting with one another. Two stickers that bind to one another may be on the same strand or on two different strands. Moreover, many strands can be connected to one another through a chain of such bonds. We call a group of m strands connected through a series of intermolecular bonds a multimer of size m , or an m -mer. There are many ways a multimer of size m can form: any combination of bonds that occur either intra- or inter-molecularly within a group of m strands, such that each strand is reachable from every other by following a series of intermolecular bonds, is an m -mer.

We consider a system of M strands present in a container of volume V , such that their concentration is $c^{\text{tot}}=M/V$. We take the thermodynamic limit of M and V going towards infinity with their ratio staying constant. We seek to predict how frequently multimers comprised of m strands form in this system, and how this frequency changes with m . We define c_m as the concentration of multimers of size m , such that

$$c^{\text{tot}} = \sum_{m=1}^{\infty} m c_m.$$

REVIEWERS' COMMENTS

Reviewer #1 (Remarks to the Author):

The authors have addressed all our comments.

Reviewer #2 (Remarks to the Author):

I have read the response to the reviews (others included). I believe that the manuscript is sufficiently improved. I am satisfied enough that I recommend publication.

Reviewer #3 (Remarks to the Author):

I am satisfied with the reply of the authors. They have clarified all my concerns.

Response to reviewers

Our submission requires “a separate point-by-point response to the reviewers’ comments, reproduced verbatim.”

Reviewer comments are reproduced verbatim in *italics* below. We thank the reviewers for their time and careful attention to our manuscript.

Reviewer #1 (Remarks to the Author):

The authors have addressed all our comments.

Reviewer #2 (Remarks to the Author):

I have read the response to the reviews (others included). I believe that the manuscript is sufficiently improved. I am satisfied enough that I recommend publication.

Reviewer #3 (Remarks to the Author):

I am satisfied with the reply of the authors. They have clarified all my concerns.